# *Flamenco* plasticity tunes somatic piRNAs and rewires isoforms, with implications for heritable transposon spread

Mai Moritoh[1], Chikara Takeuchi[2,3], Yurika Namba[1], Hidenori Nishihara[4], Shigeki Hirakata[1], Chie Owa[5], Shinichi Morishita[5], Yuka W Iwasaki[2], Mikiko C Siomi[1]

Transposons drive genome innovation, yet how they evade somatic piRNA defenses, reach the germline, and rewire host genes with limited cost remain unclear. Using comparative long-read sequencing in cultured *Drosophila* ovarian somatic cells, we show that the long terminal repeat transposon *Springer* modulates host gene expression through promoter-proximal intronic insertions at an AT-rich motif. Its 5′ long terminal repeat initiates transcription that splices into downstream host exons, expanding isoform diversity without adding coding sequence. We catalog 72 *Springer*-driven isoforms, highlighting its broad mutagenic potential. In parallel, we find that the *Flamenco* (*Flam*) uni-strand piRNA cluster, the principal source of somatic piRNAs, undergoes structural remodeling in ovarian somatic cells, replacing antisense transposon fragments with forward-oriented copies. This remodeling reshapes piRNA populations and erodes silencing of specific elements, including *Springer*. This relaxation of somatic repression may create conditions permissive for continued transposon activity, with possible broader implications for genome change. Our findings suggest that *Flam* plasticity can couple transposon activity to genome rewiring, with potential evolutionary consequences.

# Introduction

Transposons (hereafter referred to as TEs) are pervasive genomic parasites that threaten genome integrity across all domains of life (Kazazian, 2004; Feschotte, 2023). Their mobilization disrupts essential genes and, when occurring in the germline, compromise fertility and transmit deleterious mutations (Engels, 1983; Kazazian et al, 1988). Animals counter this threat with the Piwi-interacting RNA (piRNA) pathway, a small RNA system that enforces TE silencing and preserves reproductive fitness (Cox et al, 1998; Siomi et al, 2011; Czech et al, 2018; Ozata et al, 2019).

piRNAs derive from long single-stranded precursors transcribed from discrete genomic loci known as piRNA clusters (Aravin et al, 2003; Brennecke et al, 2007; Chirn et al, 2015). In *Drosophila*, these clusters fall into uni-strand and dual-strand types (Brennecke et al, 2007; Malone et al, 2009). Both act as genomic archives of fragmented TEs, generating antisense piRNAs that target active elements. Dual-strand clusters employ non-canonical RNA polymerase II (Pol II) and associated cofactors for bidirectional transcription and are restricted to germ cells (Mohn et al, 2014; Zhang et al, 2014; Andersen et al, 2017). Ovarian follicle cells lack this machinery and rely solely on uni-strand clusters, most prominently *Flamenco* (*Flam*) (Vaury et al, 1989; Pélisson et al, 1994; Prud'homme et al, 1995; Malone et al, 2009; Zanni et al, 2013). Loss of *Flam* function de-represses somatic TEs, establishing it as their master suppressor.

Because follicle cells encapsulate germ cells, TEs first encounter these somatic cells, where *Flam* captures and silences them (Song et al, 1997; Brennecke et al, 2007; Malone et al, 2009; Peccoud et al, 2017; Varoqui et al, 2025). Cross-species long-read sequencing revealed that *Flam* is variable even among closely related species, reflecting rapid evolutionary diversification (Signor et al, 2023). Some *Drosophila* species lack a syntenic *Flam* yet harbor *Flam*-like loci that repress long terminal repeat (LTR)-type TEs encoding an envelope (Env) protein, paralleling *Drosophila melanogaster* (van Lopik et al, 2023). These loci are enriched for antisense-oriented LTR-type TE insertion.

Host defenses constrain TE activity, but many TEs persist and can drive regulatory and evolutionary innovation. Realizing such benefits requires germline establishment, yet developmental barriers and robust somatic piRNA defenses sharply restrict access. How, then, do TEs evade piRNA surveillance to colonize the

---

[1]Department of Biological Sciences, Graduate School of Science, The University of Tokyo, Tokyo, Japan   [2]RIKEN Center for Integrative Medical Sciences, Yokohama, Japan   [3]Cecil H. and Ida Green Center for Reproductive Biology Sciences, University of Texas Southwestern Medical Center, Dallas, TX, USA   [4]Department of Advanced Bioscience, Faculty of Agriculture, Kindai University, Nara, Japan   [5]Department of Computational Biology and Medical Sciences, Graduate School of Frontier Sciences, The University of Tokyo, Chiba, Japan

Correspondence: siomim@bs.s.u-tokyo.ac.jp

germline? Once established, they can be vertically transmitted across generation (Senti et al, 2026).

Cultured *Drosophila* ovarian somatic cells (OSCs), composed solely of follicle cells (Saito et al, 2009), provide a tractable system to prove these dynamics. We previously discovered that the OSC gene *Lethal (3) malignant brain tumor* (*L(3)mbt*) harbors an intronic insertion of the LTR-type TE *Springer*, which drives hybrid splicing (TE–host fusion transcript) from the 5′ LTR to exon 4, generating a truncated isoform, *L(3)mbt-S* (Yamamoto-Matsuda et al, 2022). Loss of L(3)mbt de-represses piRNA amplification factors, such as *Vasa* and *Aubergine* (*Aub*), and causes female infertility (Sumiyoshi et al, 2016; Coux et al, 2018; Yamamoto-Matsuda et al, 2022). Yet OSCs tolerate *Springer*, offering an unprecedented window into TE–host interplay.

Here, comparative long-read sequencing of cultured OSCs shows that *Springer* can influence host gene expression via promoter-proximal intronic insertions that initiate transcription and drive host–TE hybrid splicing without adding coding sequence. Beyond *L(3)mbt*, we identified 71 additional *Springer*-driven isoforms, underscoring its mutagenic potential. We also find that *Flam* shows evidence of dynamic remodeling that can reshuffle TE fragments, reshape piRNA output, and potentially erode silencing of specific elements, including *Springer*. Such remodeling may modulate somatic control of TE activity across ovarian cell types, with possible implications for TE movement between them. TE mobilization in the germline can contribute to heritable variation and, over evolutionary timescales, may facilitate innovation. Together, our findings support a model in which *Flam* acts not only as a barrier to TE activity but also as a context-dependent modulator of genome rewiring. In this framework, transient TE up-regulation associated with *Flam* remodeling could, in some settings, expand isoform diversity and potentially yield evolutionary consequences.

# Results

## Intronic *Springer* initiates transcription and hybrid splicing without adding coding sequence

To investigate the mechanism underlying OSC-specific *L(3)mbt-S* isoform (Yamamoto-Matsuda et al, 2022), we generated a haplotype-resolved OSC genome by PacBio HiFi long-read sequencing and integrated the data with Micro-C XL profiles (Takeuchi et al, 2022). Contigs were assembled with HiFiasm and scaffolded with RagTag, yielding two haplotypes: one containing the canonical *L(3)mbt* gene (FlyBase; FBgn0002441) and the other carrying a *Springer* insertion in intron 3 (L3-*Springer*) (Fig 1A). We hereafter refer to the former as *Haplotype_1* (Hap_1) and the latter as *Haplotype_2* (Hap_2). For convenience, chromosomes other than 3R harboring *L(3)mbt* were also assigned to Hap_1 or Hap_2. Karyotyping confirmed that OSCs used in this study are diploid (Fig S1A).

Alignment of Hap_2 with *L(3)mbt-S* (Yamamoto-Matsuda et al, 2022) revealed a 7-nt splice donor (GTAAGTG) located 43 nt downstream of the 5′ LTR of L3-*Springer* (Fig 1B). Transcription initiates within the 5′ LTR, omitting exons 1–3 and including a 297-

nt *Springer* fragment that lacks an AUG start codon. Translation therefore begins at Met311 (or Met325) in exon 4, yielding L(3)mbt-S, an N-terminally truncated isoform relative to the full-length L(3) mbt-L (Yamamoto-Matsuda et al, 2022). Thus, L3-*Springer* elicits cryptic transcription initiation and hybrid splicing yet contributes no peptide sequence to the product. Hap_1 expresses the canonical transcript (*L(3)mbt-L*), consistent with the *D. melanogaster* reference genome (BDGP Release 6 plus ISO1 MT/dm6; GenBank assembly GCA_000001215.4) (Fig 1B).

Functionally, *L(3)mbt-S* compensates for loss of *L(3)mbt-L* in OSCs by repressing targets such as Vasa and Aub, which are piRNA amplification factors (Yamamoto-Matsuda et al, 2022), indicating that N-terminal truncation preserves core activity (Fig S1B). Notably, L(3)mbt-S resembles its vertebrate orthologs (Fig S1C), supporting the possibility that TE-driven isoform variation may have evolutionary consequences. Nevertheless, we cannot exclude the possibility that loss of the N terminus reflects a more ancestral state.

*Springer* (FlyBase; FBte0000333; 7,546 nt), an endogenous errantivirus (Touret et al, 2014), produces Gag/Pol from unspliced RNAs and Env from spliced RNAs (Fig S2A). Long-read RNA-seq from OSCs detected both unspliced and spliced *Springer* RNAs and confirmed use of the GTAAGTG donor without auxiliary *Springer* sequences (Fig S2B). Despite an intact polyadenylation signal (PAS) in the 3′ UTR (Fig S2C), polyadenylated L3-*Springer* RNAs were not detected in OSC RNA-seq. In S2 cells, however, L3-*Springer* was efficiently polyadenylated (Fig S2D), indicating that the PAS is intrinsically functional but not recognized in the OSC context. L3-*Springer* harbors a splice acceptor site (GTTACAG) (Fig S2B). Although the core CAG motif is maintained, the upstream 5′ adenine is suboptimal (Mount, 1982), likely reducing the splicing efficiency. This may represent an additional mechanism that permits *L(3)mbt-S* expression in OSCs.

A genomic reporter spanning the L3-*Springer* 5′ end through *L(3)mbt* exon 4 (FL-LS-mCh) yielded robust mCherry fluorescence and the expected junctions in OSCs (Figs 1C and S3A and B). A minimum construct (Δ-LS-mCh) retaining the GTAAGTG donor also supported splicing and mCherry fluorescence, whereas mutating this donor (Δ-LS-mCh-mut harboring CATTCAC) abrogated fluorescence (Figs 1D and E, and S3A, C, and D), demonstrating that the identified donor is necessary and sufficient for hybrid splicing.

## *Springer*-driven hybrid splicing pervasively rewires host isoforms in OSCs

The dm6 genome contains six full-length or near–full-length *Springer* copies (>7,500 nt; >99.5%) (Fig 2A and Table S1). In OSCs, we identified 219 insertions in Hap_1 and 209 in Hap_2; none matched the six dm6 loci (Fig 2B and C). Of these, 150 (Hap_1) and 147 (Hap_2) insertions were genic; 129 (Hap_1) and 126 (Hap_2) were intronic, with 64 and 57 oriented forward relative to host transcription (*Springer*-intF; Fig 2C and Table S2). Analysis of all 428 *Springer* insertions (219 in Hap_1 and 209 in Hap_2) showed a marked enrichment in intronic regions (Fig 2D), consistent with stronger negative selection against exonic disruption.

Long-read RNA-seq revealed hybrid splicing for 37 of 64 (Hap_1) and 35 of 57 (Hap_2) *Springer*-intF sites, including L3-*Springer* (Figs

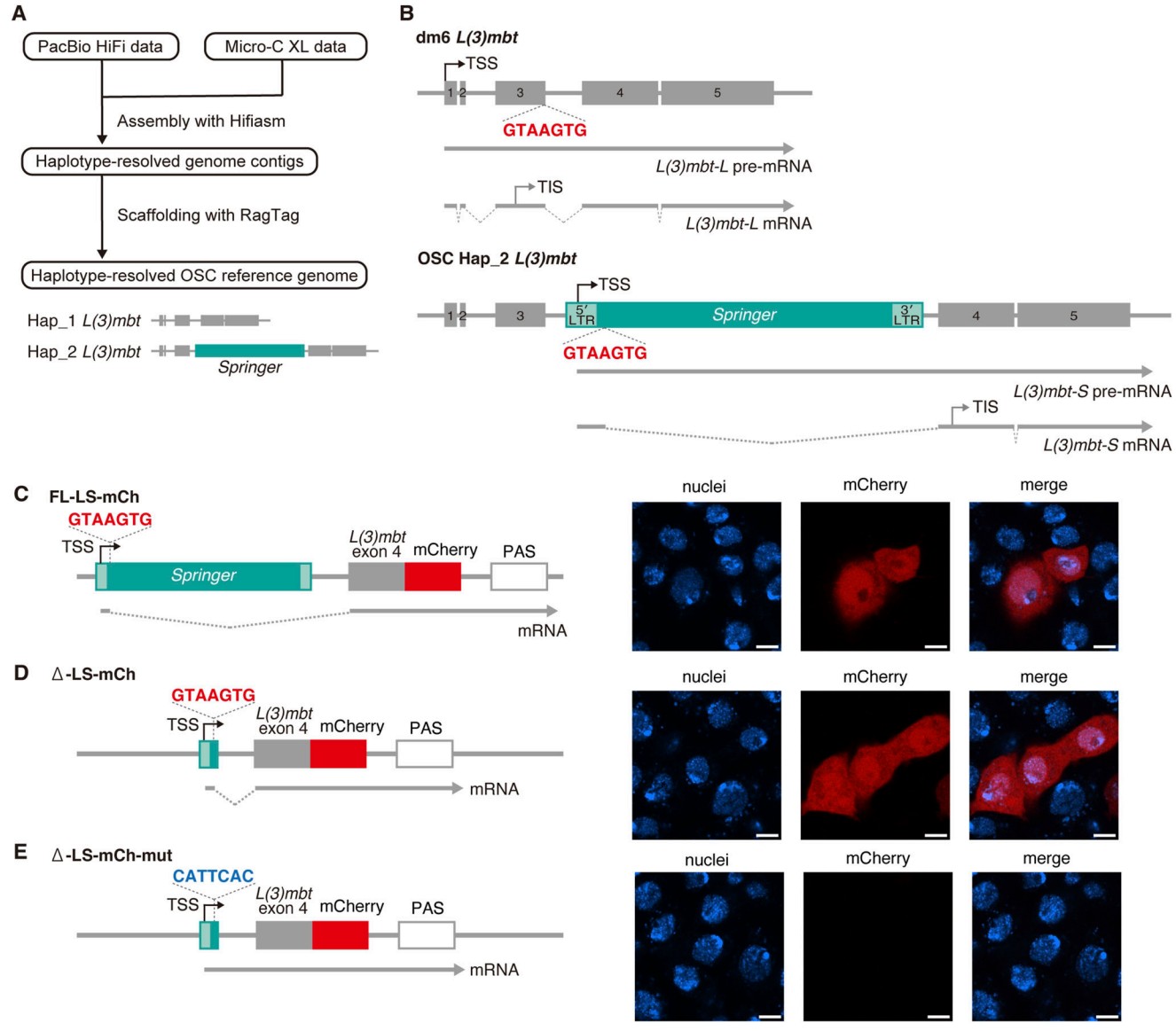

**Figure 1. L3-*Springer* introduces a 5′ splice site that drives hybrid splicing to generate the truncated *L(3)mbt-S* isoform in cultured OSCs.**
**(A)** Genome assemblies of OSC haplotypes: Hap_1 carries an intact *L(3)mbt* locus, whereas Hap_2 harbors *Springer* (L3-*Springer*) inserted within the locus.
**(B)** Structures of *L(3)mbt* in the dm6 genome (upper) and Hap_2 in the OSC genome (lower). Boxes represent exons (gray) and *Springer* (green). The L3-*Springer* 5′ splice site (GTAAGTG) is highlighted in red. Schematics of *L(3)mbt* pre-mRNAs and mRNAs are also shown. TSS, transcription start site; TIS, translation initiation site. The TSS was determined experimentally, whereas the TIS was estimated to be either Met311 or Met325 in our previous study (Yamamoto-Matsuda et al, 2022). **(C)** Plasmid-based recapitulation of *Springer*-driven hybrid splicing in OSCs. FL-LS-mCh supports splicing and mCherry expression (red). FL-LS, full-length L3-*Springer*; mCh, mCherry. Nuclei are shown in blue. Scale bar, 5 μm. TSS, transcription start site; PAS, polyadenylation signal. **(D)** Deletion of the *Springer* body (Δ-LS-mCh) preserves mCherry expression. **(E)** Mutation of the 5′ splice site (Δ-LS-mCh-mut) abolishes mCherry expression. **(D)** The splice site within Δ-LS-mCh (D) was mutated to CATTCAC (blue).

2C, E, and F and S4A; Table S3). Six genes carried *Springer*-intF at identical positions in both haplotypes (Fig 2E). In contrast, dm6 carries no *Springer*-IntF (Fig S4B) and lacks short *Springer* segments akin to Δ-LS (Fig 1D), suggesting that *Springer*-dependent isoforms are specific to the OSC genome.

Roughly 40% of *Springer*-intF insertions (27 [Hap_1] and 22 [Hap_2]/121) showed no hybrid splicing (Figs 2C and G, and S4C; Table S4). These sites lay farther from the nearest downstream exon: median 9,572 nt without splicing versus 2,661 nt with splicing (Δ = 6,911 nt; Fig 2H). We propose competition between splicing and premature polyadenylation: long downstream introns favor PAS usage within *Springer* before the acceptor is reached, yielding *Springer* mRNAs rather than hybrid isoforms. Additional parameters, such as Pol II elongation kinetics and splice acceptor strength, may also influence this balance.

### OSCs produce abundant but sense-biased *Springer* piRNAs

Although ovarian *Springer*-piRNAs effectively silence *Springer* (Sienski et al, 2012; Muerdter et al, 2013), OSCs harbor numerous

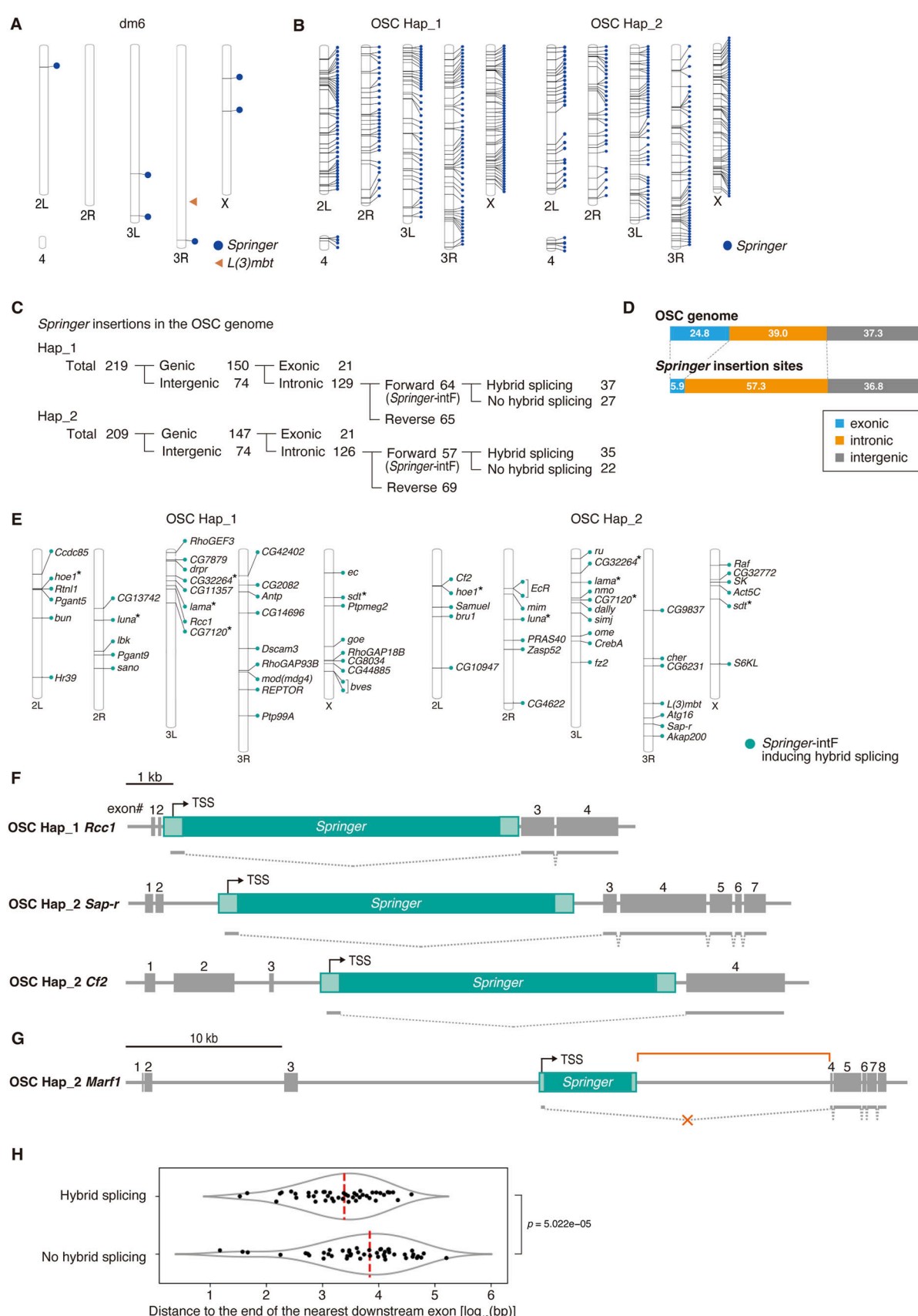

*Springer* insertions. Reanalysis of OSC piRNAs (Murano et al, 2019) showed that *Springer* piRNAs are abundant (Fig 3A), yet Piwi depletion failed to de-repress *Springer* (Fig S5A). Interestingly, *Springer* piRNAs in OSCs are strongly sense-biased compared with those produced in ovaries of the *Drosophila* ISO1 strain (Shpiz et al, 2014) (Fig 3B and C), consistent with limited silencing efficacy. *Copia*, *gypsy1*, *Transpac*, and *Xanthias* showed similar sense biases and, like *Springer*, were not markedly de-repressed in Piwi-deficient OSCs (Figs 3B and C, and S5A), indicating that residual antisense piRNAs play little role in their silencing. This is consistent with a previous report showing that antisense piRNAs in limited amounts are insufficient to trigger silencing (Post et al, 2014; Genzor et al, 2021; Ariura et al, 2024).

### *Copia* rarely drives hybrid splicing and often perturbs coding frames

*Springer* showed the strongest copy number enrichment in the OSC genome, increasing by ~36-fold relative to the dm6 reference (Fig 3D). *Copia* increased by ~3-fold, whereas *Xanthias* and *Transpac* each increased by ~2.5-fold. In contrast, *gypsy1* showed no increase (fold change < 1), for reasons that remain unclear (Fig 3B).

We identified 119 (Hap_1) and 129 (Hap_2) *copia* insertions, with 35 and 45 intronic forward-oriented sites (*copia*-intF; Fig S5B). Only 1/35 (Hap_1) and 4/45 (Hap_2) yielded detectable hybrid splicing (Fig S5C), far below *Springer*-intF frequencies. We attribute this to weaker internal donor strength (GTATGTT), competition with premature polyadenylation, and potential negative selection. Indeed, *copia*-derived exonic segments are long (1,484 nt; FlyBase; FBte0000023) and contain multiple AUGs, predisposing to frameshifts or fusion peptides that may compromise cell viability. *Xanthias*-intF and *Transpac*-intF showed no hybrid splicing (Fig S5D and E), consistent with the absence of intrinsic splicing in these elements. Genic *gypsy1* insertions were not detected (Fig S5F). Together, *Springer* uniquely and efficiently drives coding sequence–free hybrid splicing.

### *Springer* targets open, promoter-proximal introns of active genes

We investigated the genomic contexts of *Springer* insertions in OSCs (219 Hap_1; 209 in Hap_2). These insertions were preferentially located within a distinctive AT-rich repeat array and displayed a central positional bias within this array (Fig 3E). Genome-wide, insertions are enriched in introns relative to exons and intergenic regions (Fig 3F). Notably, although exonic regions comprise 23.8% of the OSC genome (Fig 2D), only 5.9% of *Springer* insertions fall within exons, substantially below the

level expected from motif frequency. Nonetheless, these data do not provide definitive evidence for a novel *Springer* integration bias; rather, they are more consistent with insertions acquired during OSC establishment and/or subsequent long-term culture.

Insertions were enriched in first/second introns and in highly expressed genes (Fig 3G and H). ATAC-seq meta-profiles centered on insertion sites revealed a sharp accessibility peak immediately upstream of integrations, absent from matched random controls (Fig 3I), indicating a preference for open chromatin near 5′ gene regions. We hypothesize that transcription initiation and local Pol II density promote *Springer* integration, consistent with known roles of host factors in TE targeting.

### Extensive *Flam* remodeling reshapes piRNA populations and silencing capacity

We compared *Flam* for *Springer*, *copia*, *Xanthias*, *Transpac*, and *gypsy1* fragments between dm6 and OSC haplotypes, focusing on inserts >300 nt because of repetitiveness. In dm6, two reverse-oriented *Springer* fragments (dm6-Sp#1 and #2) produce *Springer*-targeting piRNAs (Figs 4A and S6A). In OSCs, these are absent; instead, multiple near–full-length, forward-oriented *Springer* fragments populate *Flam* in both haplotypes (Sp#3, Sp#4, Sp#5 in Hap_1 and Sp#4 in Hap_2; Fig 4B and C), generating predominantly sense *Springer* piRNAs (Fig S6A) and explaining ineffective *Springer* silencing.

*Gypsy1* fragments also revealed remodeling: multiple reverse-oriented copies in dm6 correspond to a reconfigured set across Hap_1/Hap_2, with evidence of an ~300-kb gap relative to dm6 (Figs 4A–C and S6B). *Copia*, *Xanthias*, and *Transpac* fragments are undetectable in dm6 *Flam* (down to 20 nt searches) but present in OSC *Flam* (Tr#1 and Xa#1 in Hap_1 and co#1, Tr#1, and Xa#1/#2 in Hap_2), which produces corresponding piRNAs (Figs 3B and C, and S7A–C). Ovarian piRNAs from these TEs map mainly to dual-strand clusters (Table S5). Collectively, *Flam* undergoes pervasive remodeling that rewires piRNA populations and impacts TE activity in OSCs.

### *Flam* is a hotspot of genome rearrangement

Synteny mapping of dm6 against Hap_1 and Hap_2 revealed intact continuity at the boundaries of *Flam* but fragmented synteny across most of the locus (Fig 4D). Non-syntenic sequence comprised 52.9% (versus Hap_1) and 38.9% (versus Hap_2) of dm6 *Flam*, including a minor translocation/inversion event. Although Hap_1 and Hap_2 are broadly similar, Hap_2 uniquely harbors an

---

**Figure 2. Intronic *Springer* elements broadly promote hybrid splicing and create OSC-specific isoforms.**
**(A)** Genomic distribution of *Springer* elements ≥7.5 kb (blue dots) in the dm6 reference genome. The *L(3)mbt* locus is highlighted in orange. **(B)** Genomic distribution of *Springer* elements ≥7.5 kb (blue dots) in OSC Hap_1 and Hap_2. **(C)** Classification of *Springer* insertions in the OSC genome. **(D)** Proportions of exonic (blue), intronic (orange), and intergenic (gray) regions in the OSC genome, and the distribution of *Springer* loci across exonic (blue), intronic (orange), and intergenic (gray) regions. *Springer* insertions are enriched in introns. **(E)** Map of genes harboring *Springer*-intF that induce hybrid splicing. *, genes with *Springer* in both haplotypes, genes with two *Springer* insertions. **(F)** Examples of host genes (*Rcc1*, *Sap-r*, *Cf2*) generating *Springer*-induced isoforms. Boxes denote exons (gray) and *Springer* (green). TSS, transcription start site. **(G)** *Marf1* carries an intronic *Springer* but shows no fusion transcripts (X, orange). The orange line (top) indicates the region between the *Springer* insertion and exon 4. **(H)** Distances between *Springer* and downstream exons distinguish splicing-competent (Hybrid splicing) versus non-competent insertions (no hybrid splicing). Each dot represents a single *Springer* insertion and that the plot shows the $\log_{10}$-transformed distance (bp) to the end of the nearest downstream exon, grouped according to the presence or absence of detected hybrid splicing. Red bar, means. Mann–Whitney $U$ test, $P = 5.022 \times 10^{-5}$.

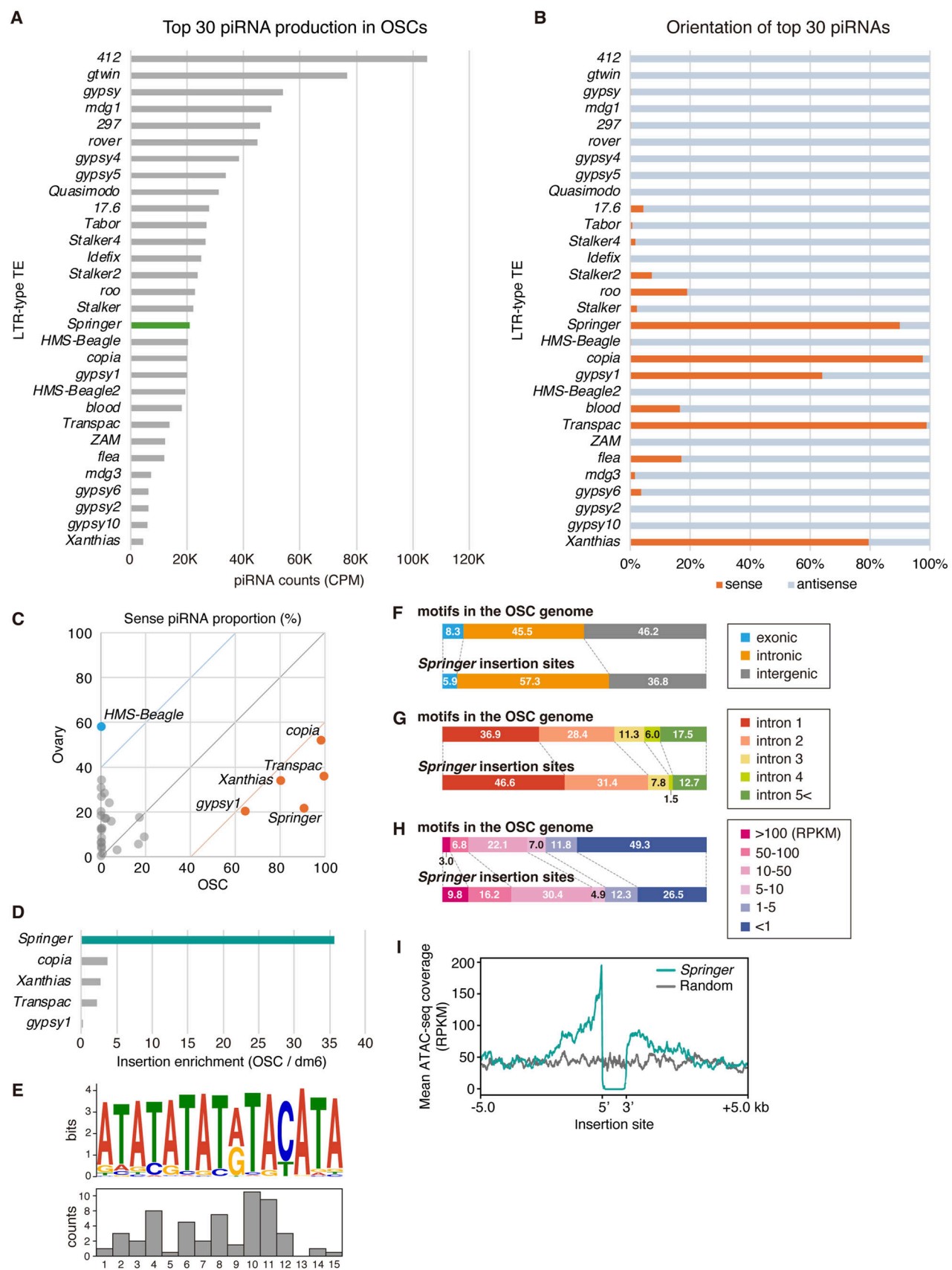

~330-kb 5′ segment densely populated with TE fragments (Figs 4D and S8A). Dot plots further showed that whereas *Flam* in dm6 and OSC (Hap_1 or Hap_2) is rich in structural variation, the remainder of the X chromosome is comparatively quiescent (Fig 5A–F): 93.5% and 92.6% of the dm6 non-*Flam* X retain synteny with Hap_1 and Hap_2, respectively.

Among major dual-strand clusters (*80F* on 3L [~44 kb, dm6], *38C* on 2L [~127 kb], *42AB* on 2R [~253 kb]) (Brennecke et al, 2007; Mohn et al, 2014) (Fig S8B), *80F* and *38C* are largely invariant, whereas *42AB* shows modest variation (Figs 5G–L and S8B–E). The tiny uni-strand cluster *20A* (~44 kb, dm6), proximal to *Flam*, seems more variant than *80F*, *38C*, and *42AB* (Figs 5M and N, and S8B and F). However, notably, *80F*, *38C*, *42AB*, and *20A* retain complete synteny between Hap_1 and Hap_2, whereas *Flam* exhibits marked haplotype disparity (Fig S8A and B), establishing *Flam* as a hotspot of genome rearrangement in OSCs.

### Low frequency of detectable structural change at *Flam* over a decade

To assess how frequently the *Flam* locus undergoes structural changes, we compared our OSC *Flam* assemblies with a previously published OSC short-read genomic dataset (Sienski et al, 2012). This comparison revealed broadly uniform read coverage across *Flam* in both haplotypes (Fig 6A). We next compared *Springer* copy number in our OSC *Flam* assemblies with two earlier OSC datasets (Sienski et al, 2012; Sytnikova et al, 2014). Relative to Sienski et al (2012), we detected only two additional *Springer* insertions over approximately the past decade, and comparison with Sytnikova et al identified only three additional insertions (Fig 6B). Together, these results suggest that cultured OSCs largely maintain their genomic and biological properties over this period despite low levels of *Springer*-targeting piRNAs and high *Springer* expression. Importantly, this stability does not undermine our central model—that *Flam* plasticity can be linked to TE activation, genomic rewiring, and potentially evolutionary consequences—because the effects of *Flam* plasticity are not limited to *Springer*.

## Discussion

Long-read, haplotype-resolved genome and RNA transcriptome analyses of cultured *Drosophila* OSCs uncovered two features that connect TE activity to host gene regulation. First, forward-oriented intronic insertions of the LTR element *Springer* can use the 5′ LTR

as a promoter to initiate transcription that splices into downstream host exons, thereby expanding host transcriptional output. Second, the somatic piRNA source locus *Flam* appears unusually susceptible to local rearrangements that can permit the reactivation of embedded TE segments. Together, these properties outline a route by which piRNA-based somatic defenses can be retuned, reshaping TE repression and, in turn, rewiring host transcriptional programs, including the emergence of new isoforms. In vivo, ovarian follicle cells are heterogeneous and persist for only a single generation during oogenesis, limiting opportunities to observe the full sequence of events from TE activation to defense retuning and transcriptome remodeling, particularly when genomic changes accrue slowly. By contrast, cultured OSCs are comparatively uniform and can be passaged for extended periods, a tractability that likely enabled us to capture these links with unusual clarity.

Importantly, somatic relaxation of TE repression driven by *Flam* plasticity could also occur in the ovary. Because de-repressed TEs in follicle cells could, in principle, gain access to neighboring germ cells, such somatic relaxation may create conditions permissive for germline entry and, ultimately, heritable change (Fig 6C). We emphasize, however, that this is a mechanistic inference based on insertion directionality and piRNA profiling in cultured OSCs. Direct measurements of germline integration rates and multigenerational transmission will be essential tests for future work, and the frequency and detectability of such events in vivo remain open questions.

Intronic TE–host hybrid transcripts arising from de novo TE insertions have been described, but prior studies have largely described isolated examples, most often detected using short-read data (Sela et al, 2007; Faulkner et al, 2009; Flemr et al, 2013; Modzelewski et al, 2021; Azad et al, 2024). Here, we systematically analyzed genome-wide de novo TE insertions that emerge in the OSC genome in association with shifts in piRNA directionality, focusing on intronic, forward-oriented events, and defined their consequences in this cell type. This approach uncovers an unrecognized mechanism that positions *Springer* as a genome-wide driver of host transcriptome remodeling and, in some cases, proteome remodeling. Consistent with this possibility, insertion of a *Springer* mini-cassette (Δ-LS) derived from Δ-LS-mCh (Fig 1D) into *Piwi* intron 2 disrupted the nuclear localization signal encoded across exons 1 and 2, converting Piwi into a predominantly cytoplasmic protein (Fig S9). Over longer timescales, such TE-driven alterations could plausibly contribute to evolutionary change.

Cultured OSCs can maintain, for extended periods, a sense-biased piRNA pool against *Springer* and other TEs, including *copia* and

**Figure 3. OSCs are enriched in sense-oriented *Springer* piRNAs and allow accumulation of *Springer* elements in active, promoter-proximal introns.**
**(A)** Counts of Piwi-bound piRNAs obtained by immunoprecipitation (Murano et al, 2019) mapped to LTR TEs (top 30) in OSCs. Green, *Springer*. **(A, B)** Strand orientation of the top 30 LTR TE-mapped piRNAs shown in (A). Orange, sense; blue, antisense. **(B, C)** Comparison of the population of sense piRNAs (%) between ovaries and OSCs for the TE-mapped piRNAs in (B). In OSCs, piRNAs from *Springer*, *copia*, *Transpac*, *gypsy1*, and *Xanthias* are sense-biased (orange), whereas those from other TEs (gray), including *HMS-Beagle* (blue), are not. **(D)** Enrichment of genomic insertions for selected LTR TEs in OSCs relative to the dm6 reference genome. For each TE family, enrichment was calculated in copy number in the OSC assembly relative to dm6. *Springer* showed the greatest enrichment in OSCs, followed by *copia*, *Xanthias*, and *Transpac*. The dm6 reference genome served as the baseline. **(E)** Sequence motif enriched at *Springer* insertion sites. The bar graph shows insertion frequency across the motif. **(E, F)** Genomic distribution of the motif in (E) across exonic (blue), intronic (orange), and intergenic (gray) regions of the OSC genome, alongside the distribution of *Springer* insertions across these regions. *Springer* insertions are biased toward introns. **(G)** *Springer* insertions preferentially occur near the 5′ ends of host genes. **(H)** *Springer* insertions preferentially occur within highly expressed genes. **(I)** Reanalysis of ATAC-seq data (Iwasaki et al, 2016) indicates that *Springer* insertions are preferentially located in open chromatin. Random, control.

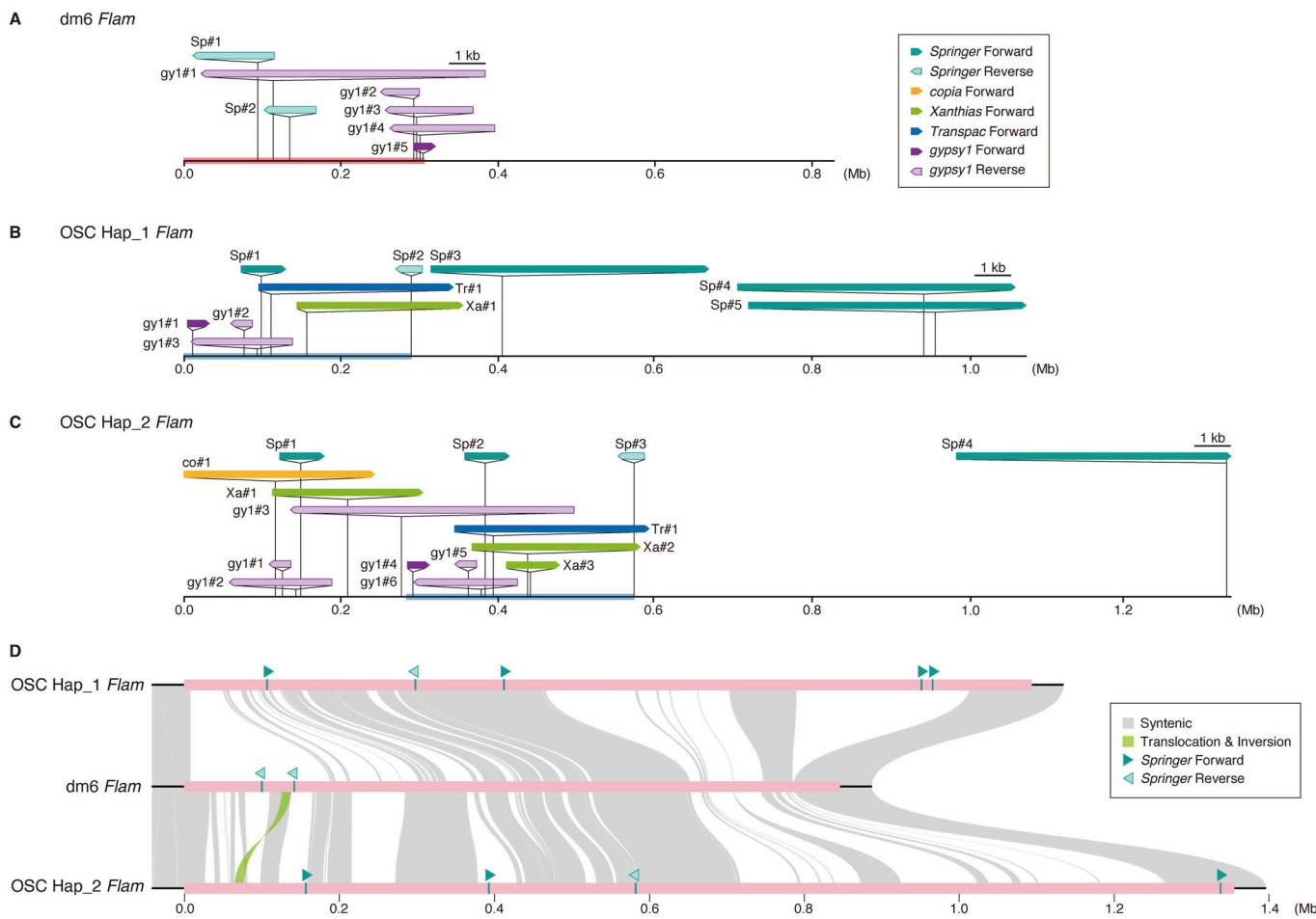

**Figure 4. The *Flam* locus exhibits extensive remodeling in cultured OSCs.**

**(A)** Organization of the *Flam* locus in the dm6 reference genome, focusing on *Springer*, *gypsy1*, *Transpac*, *copia*, and *Xanthias* fragments (>300 nt). The *Flam* locus was defined as the genomic region between the flanking genes *CG32820* (upstream) and *CG14621* (downstream) (see the Materials and Methods section for details). The 0.0 position corresponds to the 5′ boundary of the *Flam* locus. In the dm6 genome, the *Flam* locus harbors two *Springer* fragments and five *gypsy1* fragments, but no *Transpac*, *copia*, and *Xanthias* fragments. Notably, the two *Springer* fragments, Sp#1 and Sp#2 in the dm6 genome (pale green), and four *gypsy1* fragments, gy1#1–4 (pale purple), are oriented opposite (reverse) to the direction of *Flam* transcription. *Springer*-targeting piRNAs map to Sp#1 and Sp#2 (Fig S6A), indicating that these regions serve as sources of *Springer*-targeting piRNAs. The orange line represents the region absent from Hap_1 and Hap_2. **(B)** Organization of the *Flam* locus in Hap_1. *Springer* fragments Sp#1 and Sp#3–5 (green), but not Sp#2 (pale green), are oriented in the same (forward) direction as *Flam* transcription. *Transpac* (Tr#1, blue) and *Xanthias* (Xa#1, yellow green) fragments are also oriented in the same (forward) direction as *Flam* transcription. The blue line indicates the region absent from the dm6 genome; this region is identical to that in Hap_2 (see in (C)). **(C)** Organization of the *Flam* locus in Hap_2. *copia* (co#1, yellow) fragment is oriented in the same (forward) direction as *Flam* transcription. **(D)** Structural rearrangements in the *Flam* region (pink bar) and its 50-kb flanking sequences (black lines) across the dm6 genome and the OSC Hap_1 and Hap_2 assemblies. The 0.0 position corresponds to the 5′ boundary of the *Flam* locus, defined as the genomic region between *CG32820* and *CG14621*.

*Transpac*, a population unlikely to support robust silencing. We speculate that these "non-beneficial" piRNAs largely represent by-products of piRNA biogenesis and, because they impose little fitness cost under culture conditions, are subject to only weak negative selection. Notably, *Springer* itself may also carry relatively modest fitness costs: *Springer*-intF splices into host exons without adding coding sequence, expanding isoform diversity while largely decoupling transcriptome diversification from proteome expansion. This limited proteomic burden may help explain why cultured OSCs can tolerate high *Springer* copy numbers and sustain long-term coexistence.

Key open questions include the molecular basis of *Flam* remodeling, the host factors that shape *Springer*'s insertion, and the extent to which similar coding sequence–free hybrid splicing

operates across TE families, strains, and species. Resolving these issues will clarify how genomes balance defense with innovation, not by simply turning TE activity off, but by shaping when, where, and how it is allowed to matter.

# Materials and Methods

## Cell culture

OSCs (Saito et al, 2009) (DGRC Stock 288; https://dgrc.bio.indiana.edu//stock/288; RRID:CVCL_IY73) were cultured at 26°C in Shields and Sang M3 Insect Medium with L-Glutamine

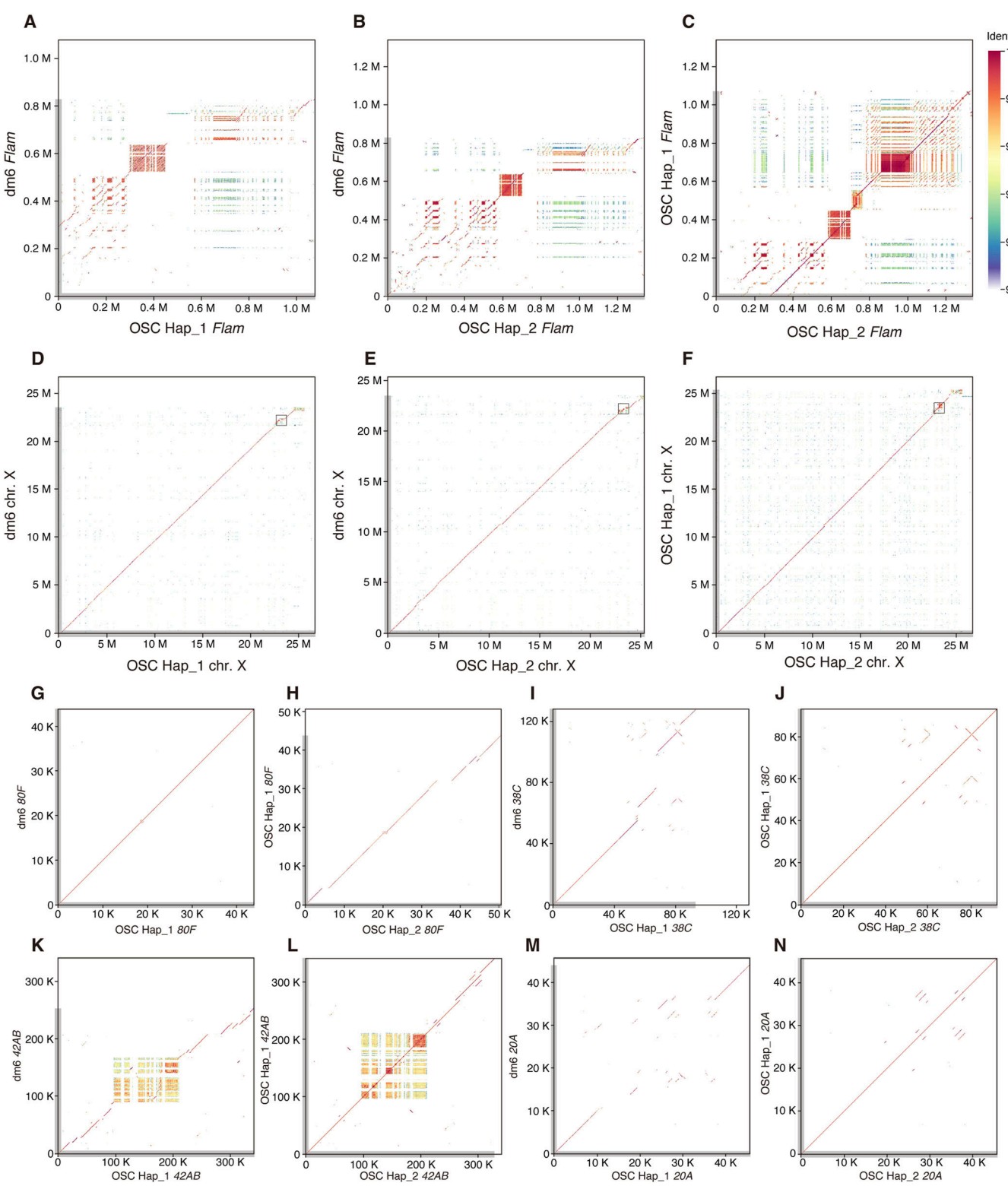

**Figure 5. The *Flam* locus is a hotspot of genome rearrangement.**
**(A)** Dot plot showing conservation of the *Flam* locus between dm6 and Hap_1, revealing large-scale rearrangements. **(B)** Conservation of the *Flam* locus between dm6 and Hap_2. **(C)** Conservation of the *Flam* locus between Hap_1 and Hap_2. **(D)** Dot plot showing conservation of the X chromosome between dm6 and Hap_1, with limited rearrangements relative to *Flam* (small open box), underscoring *Flam*'s exceptional plasticity. **(A)** Enlarged view of the boxed region is shown in (A). **(E)** Conservation of the X chromosome between dm6 and Hap_2. **(F)** Conservation of the X chromosome between Hap_1 and Hap_2. **(G)** Conservation of dual-strand cluster *80F* between dm6 and Hap_1. **(H)** Conservation of *80F* between Hap_1 and Hap_2. **(I)** Conservation of dual-strand cluster *38C* between dm6 and Hap_1. **(J)** Conservation of *38C* between Hap_1 and Hap_2. **(K)** Conservation of dual-strand cluster *42AB* between dm6 and Hap_1. Although patches of reduced conservation are

(US Biological) supplemented with 10% fly extract (Saito et al, 2009), 10% FBS (NICHIREI), 10 mU/ml insulin, and 0.6 mg/ml glutathione. S2 cells were cultured at 26°C in Schneider's *Drosophila* Medium (Gibco) with 10% FBS.

### Genome sequencing

For HiFi sequencing, genomic DNA was extracted from OSC using a Genomic-tip 20/G (QIAGEN) and a Genomic DNA Buffer Set (QIAGEN). To assess the quality and integrity of the genomic DNA samples, we ran them for sequencing on Pippin Pulse pulsed-field electrophoresis (Sage Science). We used 0.75% agarose gel (SeaKem Gold; Lonza) on a pre-set default program of 5–80 kb (75 V, 16 h run time) with ~500 ng DNA/lane. According to the PN101-853-100 protocol (PacBio), the genomic DNA was sheared using a Megaruptor 2 (Diagenode), and the fragments were collected using 0.45× AMPure PB beads (PacBio). The size of the DNA fragments was checked using a TapeStation System (Agilent Technologies). SMRTbell libraries were prepared using an SMRTbell Express Template Prep Kit 2.0 (PacBio) and a BluePippin DNA Size Selection System (Sage Science). The selected libraries were bound with Sequencing Primer v2 and Sequel II Polymerase 2.0 using a Sequel II Binding Kit 2.0 (PacBio). Sequencing was performed on a Sequel II instrument (PacBio) in CCS mode using an SMRT Cell 8 M and a Sequencing Plate 2.0. This process included 8 h of pre-extension and 30 h of movie viewing.

### Assembly of the OSC genome

A haplotype-resolved genome of the OSC cell line was assembled using the HiFiasm software (Cheng et al, 2021). This assembly was performed using PacBio long-read sequencing data generated in this study, in conjunction with publicly available Micro-C XL data from OSC cells (Takeuchi et al, 2022). The resulting contigs were subsequently aligned and scaffolded against dm6 reference genome with the Ragtag toolkit (Alonge et al, 2022).

### Chromosome preparation and DAPI staining

OSCs were treated with 50 $\mu$g/ml colchicine to inhibit microtubule polymerization. Cells were incubated at 26°C for 1 h, washed once with PBS (without $Ca^{2+}$ and $Mg^{2+}$), and collected into 1.5-ml microcentrifuge tubes. After removal of the supernatant, the cells were resuspended in 500 $\mu$l of 75 mM KCl prepared in distilled water and incubated at 26°C for 30 min to induce hypotonic swelling. Carnoy's fixative (methanol:acetic acid = 3:1) was added dropwise (500 $\mu$l) to the cell suspension, followed by centrifugation. Cells were washed three times with Carnoy's fixative and then incubated in 500 $\mu$l of fresh fixative at 4°C for 1 h. Fixed cells were mounted with VECTASHIELD Mounting Medium with DAPI (Vector Laboratories). Chromosome spreads were imaged using a confocal laser scanning microscope (LSM 980; Carl Zeiss) equipped with a

Plan-Apochromat 63×/1.4 Oil DIC M27 objective lens. DAPI was excited with a 405-nm laser.

### OSC gene annotation

To annotate the OSC genome, gene annotations were transferred from the dm6 reference genome. First, we generated chain files to map coordinates between the dm6 reference and each OSC haplotype (Hap_1 and Hap_2) using the flo (https://github.com/wurmlab/flo) (Pracana et al, 2017). Using these chain files, we then transferred the dm6 gene annotations (UCSC) to our Hap_1 and Hap_2 assemblies with the liftOver utility from the UCSC toolkit (Hinrichs et al, 2006).

### RNAi and transfection

For RNAi, OSCs were first electroporated with 200 pmol of siRNA in 20 $\mu$l of Solution SF (Cell Line Nucleofector Kit SF; Lonza) using a Nucleofector 96-well Shuttle device (Lonza). After electroporation, the cells were left at RT for 10 min before plating into culture dishes, and then incubated at 26°C for 48 h. The cells were subsequently subjected to a second electroporation with 600 pmol of siRNA dissolved in 100 $\mu$l of transfection buffer (180 mM sodium phosphate, pH 7.2, 5 mM KCl, 15 mM $MgCl_2$, and 50 mM D-mannitol) using the Nucleofector 2b system (Lonza). After a 10-min rest at RT, the cells were plated again and incubated at 26°C for an additional 48 h before collection. siRNA sequences are listed in Table S6. For plasmid transfection, $1.0 \times 10^7$ OSCs or S2 cells were mixed with 6–15 $\mu$g of plasmid DNA in 100 $\mu$l of the same transfection buffer modified from a previous report (Nye et al, 2014) and electroporated with a Nucleofector 2b device (Lonza). Cells were maintained at 26°C and harvested 48 h after transfection.

### Nuclear fractionation

Nuclear fractionation was carried out following a modified version of a previously described protocol (Onishi et al, 2020). Briefly, the cells were suspended in hypotonic buffer (10 mM HEPES-KOH [pH 7.3], 10 mM KCl, 1.5 mM $MgCl_2$, 0.5 mM DTT, 2 $\mu$g/ml leupeptin, 2 $\mu$g/ml pepstatin A, and 0.5% aprotinin), gently mixed by pipetting, and disrupted by repeated passage through a 25-gauge needle. Lysates were centrifuged at 400$g$ for 10 min, and the pellet was collected as the crude nuclear fraction, while the supernatant was retained as the cytoplasmic fraction. The cytoplasmic fraction was adjusted to 200 mM KCl and clarified by centrifugation at 20,000$g$ for 20 min. The nuclear pellet was washed with hypotonic buffer, resuspended in chromatin co-immunoprecipitation buffer (50 mM HEPES-KOH [pH 7.3], 200 mM KCl, 1 mM EDTA, 1% Triton X-100, 0.1% sodium deoxycholate, 2 $\mu$g/ml leupeptin, 2 $\mu$g/ml pepstatin A, and 0.5% aprotinin), then sonicated and centrifuged at 20,000$g$ for 20 min. The resulting supernatant was used as the nuclear extract for Western blotting.

---

apparent in the central portion, a continuous diagonal is detected across the entire plot, indicating a high degree of overall conservation. **(L)** Conservation of *42AB* between Hap_1 and Hap_2. **(M)** Conservation of uni-strand cluster *20A* between dm6 and Hap_1. **(N)** Conservation of *20A* between Hap_1 and Hap_2.

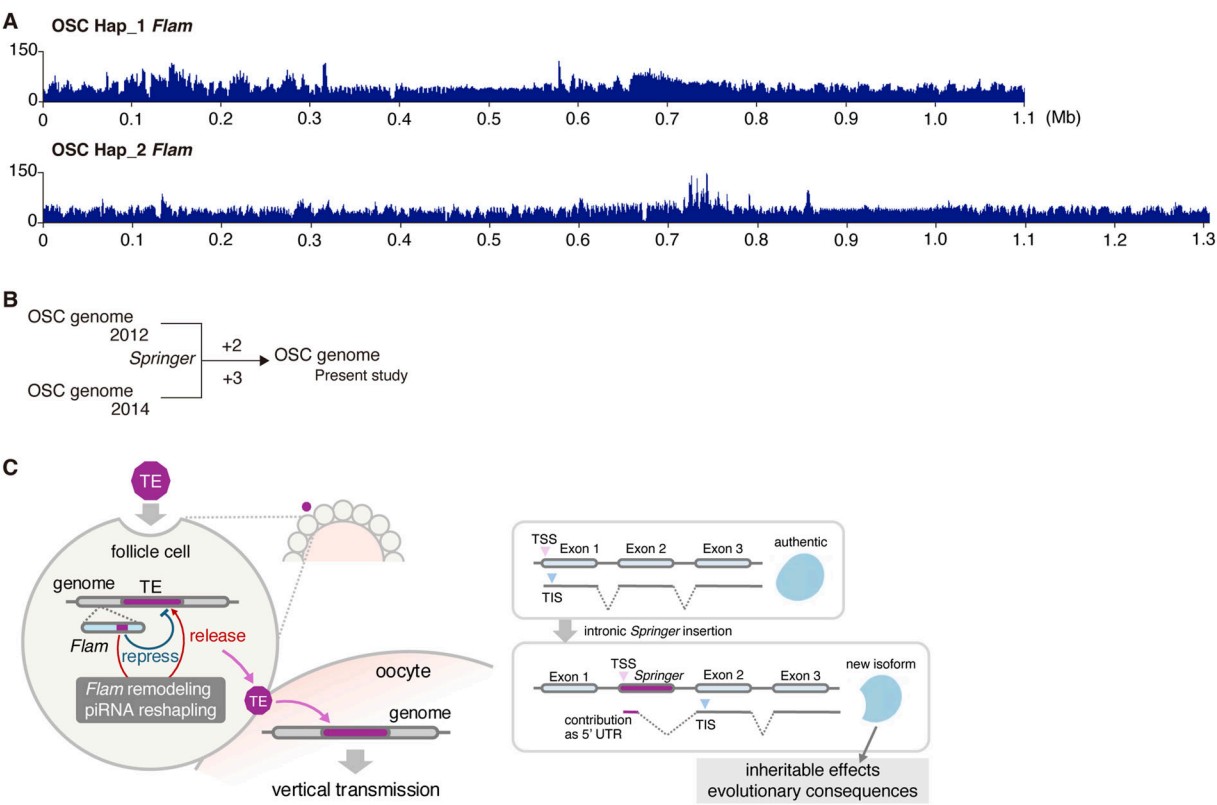

**Figure 6. *Flam* remains structurally robust for at least a decade after OSC establishment.**
**(A)** Comparison of the *Flam* locus in the OSC Hap_1 and Hap_2 assemblies (this study) with a previously published OSC genomic short-read dataset (Sienski et al, 2012). Reads were mapped to the Hap_1 and Hap_2 assemblies. The y-axis shows normalized genomic sequencing read coverage in OSCs across the *Flam* locus, indicating that this region has remained structural uniform over at least a decade. The x-axis represents the *Flam* locus (Mb). **(B)** Comparison of *Springer* insertions in the OSC Hap_1 and Hap_2 assemblies with earlier OSC genome analyses. "+2" indicates that two additional *Springer* insertions were detected in OSCs (this study) relative to Sienski et al (2012), and "+3" indicates that three additional insertions were detected relative to Sytnikova et al (2014). **(C)** A proposed model for how *Flam* plasticity shapes host–TE interactions. Left, *Flam* incorporates fragments of TEs that enter ovarian follicle (somatic) cells and generates piRNAs that repress cognate elements. *Flam* structural plasticity may also reshape piRNA populations and weaken silencing of specific TEs, potentially making germline entry and vertical transmission more permissive. Right, intronic *Springer*-driven hybrid splicing may rewire host isoforms, with possible heritable effects and evolutionary consequences.

## Western blotting

Proteins from nuclear extracts were resolved by SDS–polyacrylamide gel electrophoresis and transferred onto Immobilon-P membranes (Millipore). Membranes were blocked in PBS containing 0.1% Tween-20 (T-PBS) supplemented with 5% skim milk and then incubated with anti-L(3)mbt antibody (1:1,000 dilution) (Yamamoto-Matsuda et al, 2022). After washing with T-PBS, membranes were incubated with horseradish peroxidase–conjugated anti-mouse IgG secondary antibody (1:10,000 dilution; cat. no. 55558; Cappel). Protein bands were visualized using Clarity Western ECL Substrate (Bio-Rad), and chemiluminescence signals were captured with a ChemiDoc XRS Plus imaging system (Bio-Rad).

## RT-qPCR

Total RNA was extracted from OSCs using ISOGEN II (NIPPON GENE) and treated with TURBO DNase at a final concentration of 0.04 U/µl (Thermo Fisher Scientific) to remove residual genomic DNA. Reverse transcription was performed with ReverTra Ace (Toyobo) according to the manufacturer's protocol. Quantitative PCR (qPCR) was carried out on a StepOnePlus Real-Time PCR System (Thermo Fisher Scientific) using THUNDERBIRD Next SYBR qPCR Mix (Toyobo). Relative transcript abundance was determined using the ΔΔCt method, with *rp49* as the internal reference gene. Primer sequences are listed in Table S6.

## Iso-seq and bioinformatic analysis

Total RNA was extracted from cultured cells using ISOGEN II (NIPPON GENE), followed by DNase treatment, phenol/chloroform purification, and ethanol precipitation. Full-length cDNA libraries were prepared with the SMRTbell Prep Kit 3.0 (Pacific Biosciences) according to the manufacturer's protocol. Sequencing was performed on a PacBio Sequel II platform in HiFi read mode (1-cell run), generating 4,504,293 HiFi reads comprising 9.73 Gb of sequence data, with a read N50 of 2,234 bp, an average read length of 2,160 bp, and an average read quality of Q45. HiFi reads were processed using the Iso-Seq3 pipeline (v.4.0.0) (https://github.com/ylipacbio/IsoSeq3). Demultiplexing and primer removal were conducted with lima in Iso-Seq mode, specifying custom 5′ and 3′ primer sequences (5′-GCAATGAAGTCGCAGGGTTGG

G-3′ and 3′-GTACTCTGCGTTGATACCACTGCTT-5′). Full-length non-concatemer (FLNC) reads were refined with the --require-polya option to confirm poly(A) tails, and high-quality consensus isoforms were obtained by clustering with quality value scoring enabled. The curated isoforms were aligned independently to OSC genome Hap_1 and Hap_2 using minimap2 (v.2.24-r1122) (Li, 2018) in splice-aware mode with quality value support, while suppressing secondary alignments. Alignment files were processed with SAMtools (v.1.16.1) (Danecek et al, 2021) to convert, sort, and index BAM files for downstream analysis.

### Plasmid construction

For the L3-*Springer* plasmid (Fig S2D), the L3-*Springer* sequence from the OSC Hap_2 genome was PCR-amplified from OSC genomic DNA and inserted into a modified pAc-based vector lacking the promoter region using Mighty Cloning (TaKaRa Bio). For FL-LS-mCh (Fig 1C), a genomic fragment spanning the *L(3) mbt* locus from the L(3)-*Springer* insertion site in intron 3 to the putative start codon of exon 4 (OSC Hap_2 genome) was PCR-amplified from OSC genomic DNA. This fragment, together with an mCherry sequence PCR-amplified from pIB_DDX4N_mCherry_CRY2PHR, was assembled into a modified pAc-based expression vector lacking the promoter using NEBuilder HiFi DNA Assembly Master Mix (New England Biolabs). The Δ-LS-mCh (Fig 1D) and Δ-LS-mCh-mut (Fig 1E) constructs were generated by inverse PCR. For the Piwi plasmid (Fig S9), the *Piwi* genomic region (including introns) was PCR-amplified from OSC genomic DNA and assembled into the same modified vector with a C-terminal FLAG tag using NEBuilder HiFi DNA Assembly Master Mix (New England Biolabs). To generate the Piwi+Δ-LS construct (Fig S9), the Δ-LS fragment was amplified from the Δ-LS-mCh template and incorporated into the Piwi plasmid using NEBuilder HiFi DNA Assembly Master Mix. Primer sequences used in these plasmid constructions are listed in Table S6.

### 3′ RACE

S2 cells were transfected with the L3-*Springer* plasmid by electroporation using the Nucleofector 2b device (Lonza) with program N-20 and cultured at 26°C for 2 d. Total RNA was extracted with ISOGEN II (NIPPON GENE), followed by DNase treatment and phenol/chloroform extraction with ethanol precipitation. 3′ RACE-ready cDNA was synthesized using the SMARTer RACE 5′/3′ Kit (TaKaRa Bio) according to the manufacturer's instructions. The resulting 20 $\mu$l cDNA was diluted with 90 $\mu$l Tricine-EDTA buffer. PCR amplification was carried out with SeqAmp DNA Polymerase (TaKaRa Bio) using a gene-specific primer containing a 15-bp overlap sequence for cloning (5′-gattacgccaagcttGCTCCATCTTCAGTGAGCTCGTGTGCGC-3′) under touchdown PCR conditions (Program 1). PCR products were resolved on a 1% TBE agarose gel, and the target band was excised and purified with the FastGene Gel/PCR Extraction Kit (Nippon Genetics). Purified fragments were assembled into the pRACE vector using NEBuilder HiFi DNA Assembly Master Mix (New England Biolabs) and transformed into NEB 5-alpha

Competent *E. coli* (New England Biolabs). Colony PCR was performed with SapphireAmp Fast PCR Master Mix (TaKaRa Bio) using the gene-specific primer and M13 forward primer. PCR products were treated with Exonuclease I (New England Biolabs) and Shrimp Alkaline Phosphatase (New England Biolabs) and sequenced using the BigDye Terminator v3.1 Cycle Sequencing Kit (Thermo Fisher Scientific).

### RT–PCR

Total RNA was extracted from OSCs using ISOGEN II (NIPPON GENE) and treated with TURBO DNase at a final concentration of 0.04 U/$\mu$l (Thermo Fisher Scientific) to eliminate residual genomic DNA. Reverse transcription was performed with the Transcriptor First Strand cDNA Synthesis Kit (Roche) using oligo(dT) primers. The resulting cDNA was amplified with Q5 High-Fidelity DNA Polymerase (New England Biolabs). PCR products were resolved on a 1% TBE agarose gel, stained with SYBR Gold (Thermo Fisher Scientific) for 10 min, and visualized with a transilluminator (Fig S3A). Primer sequences are listed in Table S6.

### Live-cell imaging

OSCs were cultured on glass-bottom dishes (MATSUNAMI). Hoechst 33342 (Lonza) was added to the medium at a final concentration of 1 $\mu$g/ml, and cells were incubated at 26°C for 30 min. Live-cell imaging was performed as previously described (Yamazaki et al, 2023), using a confocal laser scanning microscope (LSM 980; Carl Zeiss) equipped with a Plan-Apochromat 63×/1.4 Oil DIC M27 objective lens at RT. Hoechst 33342 and mCherry were excited with 405 and 561 nm lasers, respectively, and images were acquired under identical acquisition settings across all samples within each experiment.

### *Springer* insertion analysis

The dm6 and OSC genomes were analyzed with RepeatMasker (v.4.1.5) (Smit et al, 2015) to annotate TEs. From the resulting GFF files, entries corresponding to the LTR TE *Springer* were extracted. Insertions ≥7,500 nt in length (≥99.5% of the reference consensus length) were classified as full-length *Springer* elements. Genomic insertion sites of these elements were visualized using Phenogram (https://visualization.ritchielab.org/phenograms/plot; Fig 2A–C) (Wolfe et al, 2013). Phenogram was also used to visualize selected insertions in Fig 2E.

### *Springer* insertion site analysis

We categorized novel *Springer* insertions based on their location relative to the lifted-over gene annotations. Insertions were classified as "genic" if they fell within an annotated exon or intron, and "intergenic" otherwise. Genic insertions were further subdivided into "exonic" or "intronic." To analyze the distance to the nearest splice site, we focused on intronic insertions located on the sense strand relative to the host gene and calculated the distance from the insertion site to the start of the nearest downstream exon using a custom script. Insertions that were

found within transcripts identified by our Iso-seq data, potentially creating TE-gene fusion transcripts, were classified as "hybrid splicing" events (see the Data and Materials Availability section for details). For the analysis in Fig 3E–H, we annotated the genomic features of both the FIMO-identified motif sites and the novel *Springer* insertion sites by mapping their coordinates to the dm6 reference genome. Feature distribution was determined using annotatePeaks.pl. 3′ UTR, 5′ UTR, exon, and noncoding were categorized as "exonic," whereas intergenic, TTS, promoter-TSS were grouped as "intergenic." To assess the expression level of genes targeted by insertions or containing the motif, we used OSC gene expression data from FlyBase (release FB2025_01). The intron number (e.g., first intron, second intron) for each intronic insertion was also determined from the HOMER output (Heinz et al, 2010).

### TE insertion site and motif analyses

To identify *Springer* and *copia* insertion sites in the dm6 genome, 100-bp flanking sequences of these insertions from the OSC genome were mapped back to the dm6 reference. The center points of successfully mapped flanking pairs were defined as the insertion site. Annotations of these insertion sites are analyzed using Homer (annotatePeaks.pl) (Heinz et al, 2010). De novo motif discovery on these insertion sites was performed using MEME-ChIP (Machanick & Bailey, 2011). *Springer* motif sites were defined by FIMO (Grant et al, 2011) at the threshold of $P = 0.001$ and annotated in the same way.

### Small RNA analysis

Previously published small RNA-seq datasets were re-analyzed: ovary small RNAs from the *D. melanogaster* ISO1 strain *y[1]; cn[1] bw[1] sp[1]* (SRR827770) (Shpiz et al, 2014) and Piwi-bound small RNA-seq from OSCs (SRR9158321) (Murano et al, 2019). For the ovary dataset, the Illumina 3′ adapter sequences (TGGAATTCTCGGGTGCCAAGGAACTCCAGTCAC) were removed using Cutadapt (v.4.6) (Martin, 2011). Reads outside the 20–35 nt size range were discarded, leaving ~13 million reads for analysis. For the OSC dataset, preprocessing was performed following the previous study (Murano et al, 2019), in which four random nucleotides from each end and adapter sequences were removed using Cutadapt, and only reads between 20–35 nt size range were retained for analysis, resulting in ~3.7 million reads. Trimmed ovary reads were aligned to dm6, whereas OSC reads were mapped to a merged OSC genome assembly (Hap_1 and Hap_2). In both cases, alignments were performed using Bowtie (v.1.3.1) (Langmead et al, 2009) with identical parameters (-v 1, ≤1 mismatch; --best), ensuring consistency between datasets. Only reads that successfully mapped to the dm6 or merged OSC genome were retained for subsequent analyses. For piRNA production analysis (Fig 3A–C), genome-mapped reads were further aligned to consensus sequences of *Drosophila* TEs obtained from FlyBase using Bowtie with no mismatches allowed, and random multi-mapping was performed with the -k 1 option. Read counts were normalized to counts per million mapped reads (CPM). To quantify strand-specific mapping,

custom Python scripts were used to count the number of sense and antisense reads for each TE, based on SAM alignment flags. For TE locus-specific analyses (Figs S6 and S7), the previously generated alignment files were converted to BAM format, sorted, and indexed using SAMtools. Normalized coverage tracks were then produced with deepTools (bamCoverage, v3.5.5) (Ramírez et al, 2016) using counts per million mapped reads (CPM) normalization. The resulting BigWig files were visualized in Integrative Genomics Viewer (Robinson et al, 2011).

### TE enrichment analysis

The dm6 and the merged OSC genome assembly (Hap_1 + Hap_2) were annotated with RepeatMasker. From the resulting GFF files, insertions ≥99.5% of the TE consensus length were defined as full-length. For each TE, genomic intervals were merged, the occupied bases were summed, and genome occupancy was calculated as the fraction of occupied bases relative to total genome size. Enrichment was defined as the ratio of occupancy in OSC genome to that in dm6. Custom Python scripts (Biopython-based) were used for these calculations (Fig 3D).

### ATAC-seq analysis

Previously published ATAC-seq data (Iwasaki et al, 2016) were re-analyzed for this study. First, raw reads were processed for quality control: read errors were corrected using Rcorrector (v1.0.4) (Song & Florea, 2015) and TranscriptomeAssemblyTools (https://github.com/harvardinformatics/TranscriptomeAssemblyTools), and adapter sequences were removed using TrimGalore (v0.6.6) (https://github.com/FelixKrueger/TrimGalore). Reads shorter than 50 bp were subsequently discarded. The filtered reads were aligned to each of the Hap_1 and Hap_2 haplotypes of the OSC genome using Bowtie2 (v2.5.4) (Langmead & Salzberg, 2012) with the following parameters: "--maxins 4000 --very-sensitive --no-mixed --no-discordant." Uniquely mapped reads were selected using SAMtools (v1.21) (Danecek et al, 2021). Read coverage was then calculated and normalized to reads per kilobase per million mapped reads (RPKM) using the bamCoverage tool from the deepTools suite (v 3.5.4) (Ramírez et al, 2016), setting the bin size to 10 bp and the minimum mapping quality to 10. For datasets with biological replicates, the signal was averaged using the bigwigAverage tool from deepTools to generate a single representative BigWig file per haplotype. These files were used to compute signal distributions centered on *Springer* insertion sites (Hap_1: 219 sites; Hap_2: 209 sites) using the computeMatrix tool from deepTools in scale-regions mode. The signal was calculated across each 10-kb window (5 kb upstream and downstream of each site), which was divided into 10-bp bins, with missing data treated as zero. As a control, 200 genomic windows (7,546 bp each, matching the length of a full-length *Springer* element) were randomly selected using the BEDTools random command (v2.31.1) (Quinlan & Hall, 2010). These control regions were processed through the same computeMatrix pipeline. The resulting signal matrices for both *Springer* and control sites were used to generate mean profile plots with the plotProfile tool from deepTools. The signal profiles for Hap_1 and Hap_2 were then averaged and plotted using R (R Core Team, 2025).

### Definition of piRNA cluster regions

The *Flam* locus was defined as the genomic region between *CG32820* and *CG14621*, which are conserved flanking genes identified in both the dm6 annotation and the lifted-over OSC gene annotations. Other piRNA cluster regions were defined based on the *D. melanogaster* piRNA cluster annotations from the ProTRAC database (Rosenkranz & Zischler, 2012), using sequence coordinates corresponding to the reported cluster boundaries. See Table S7 for locus information.

### *Flam* structural rearrangement analysis

Structural differences among the dm6 and OSC genome assemblies were analyzed using minimap2 (Li, 2018) and SyRI (Goel et al, 2019). Genome alignments were performed with minimap2 (v2.29) using the "-cx asm5 –eqx" option in two runs: OSC Hap_2 assembly was aligned to dm6, and dm6 was aligned to the OSC Hap_1 reference assembly. The resulting PAF files were analyzed with SyRI (v1.7.1) using the "--cigar –nosnp" option. Rearrangements in the *Flam* region and its 50-kb flanking sequences were visualized by excluding highly diverged regions, using plotsr (v1.1.1) (Goel & Schneeberger, 2022) with the "—itx" mode.

### Dot plot analysis

Dot plots were generated with ModDotPlot (Sweeten et al, 2024) to compare, for each genomic region, the following pairs: dm6 versus OSC Hap_1, dm6 versus OSC Hap_2, and OSC Hap_1 versus OSC Hap_2. Unless noted otherwise, default settings were used with an identity threshold of -id 90. The sliding window parameter -r was set per analysis as follows: 600 for the *Flam* locus (Fig 5A–C), 1,000 for the X chromosomes (Fig 5D–F), 800 for *80F* (Figs 5G and H, and S8C), 600 for *38C* (Figs 5I and J, and S8D), 600 for *42AB* (Figs 5K and L, and S8E), and 800 for *20A* (Figs 5M and N, and S8F).

### Genome comparison analysis

DNA-Seq data from previous study (Sienski et al, 2012) were mapped to the haploid genome (Hap_1/Hap_2) using Bowtie2 (Langmead & Salzberg, 2012) with default settings. A TE insertion was considered preexisting if reads spanning the TE and its flanking sequences were found in the data. Conversely, TE insertions that lacked support from such spanning reads were identified as novel insertions.

### Identification of insertion sites in other OSC genomes

The genomic DNA-Seq datasets from previous studies (Sienski et al, 2012; Sytnikova et al, 2014) are mapped to Hap_1 and Hap_2, respectively, using Bowtie2. Then, if the reads which intersect TE and flanking sequencings are detected, these TEs are considered as existing in the genome of the other dataset.

### Immunofluorescence

OSC cells were fixed and processed for immunostaining as described previously (Saito et al, 2010). Cells were incubated with an anti-FLAG monoclonal antibody (M2, 1:1,000 dilution; Sigma-Aldrich) as the primary antibody, followed by incubation with an Alexa Fluor 488–conjugated anti-mouse IgG secondary antibody (1:500 dilution; Thermo Fisher Scientific). After washing, samples were mounted in VECTASHIELD Mounting Medium with DAPI (Vector Laboratories) and imaged using a confocal laser scanning microscope (LSM 980; Carl Zeiss).

### Statistics and reproducibility

RT-qPCR (Figs S1B and S5A) was performed over three independent experiments, and other live-cell imaging (Fig 1C–E), Western blotting (Fig 1B), 3' RACE (Fig S2D), RT–PCR (Fig S3A), and immunofluorescence (Fig S9) were performed over two independent experiments. Statistical procedures are described in the figure legends. Sample sizes were not predetermined, and experiments were performed without randomization or blinding.

## Data and Materials Availability

All sequencing data supporting this study are available at NCBI. This Whole Genome Shotgun project has been deposited at DDBJ/ENA/GenBank under accession numbers JBVFKA000000000 and JBVFKB000000000. The versions described in this article are JBVFKA010000000 and JBVFKB010000000. Raw genome HiFi reads and Iso-Seq reads are available in the Sequence Read Archive (SRA) under accession numbers SRR37248010 and SRR37247793 within BioProject PRJNA1417524. Code and data are available at GitHub (https://github.com/Chikara-Takeuchi/2025_moritoh_TE/tree/master).

## Supplementary Information

## Acknowledgements

We thank H Yamazaki, S Yamanaka, R Saito, K Saito, and other members of the Siomi laboratory at the University of Tokyo for useful discussions. We thank I Oka and T Morita for technical assistances. We dedicate this work to the memory of Greg Hannon, a pioneering leader in this field, whose contributions continue to inspire researchers worldwide. Japan Society for the Promotion of Science 24K18093 (Y Namba), Japan Society for the Promotion of Science 25H01308 (H Nishihara), Japan Society for the Promotion of Science 24K09376 (S Hirakata), Japan Society for the Promotion of Science 25H01305 (YW Iwasaki), Japan Society for the Promotion of Science 25H01303 (MC Siomi), Japan Society for the Promotion of Science Research Fellowship for Young Scientists (DC2) 22KJ2696 (C Takeuchi), and Fusion Oriented REsearch for disruptive Science and Technology JPMJFR224L (YW Iwasaki).

# Life Science Alliance

## Author Contributions

M Moritoh: data curation, formal analysis, investigation, visualization, and writing—original draft, review, and editing.
C Takeuchi: data curation, formal analysis, and investigation.
Y Namba: formal analysis, validation, and investigation.
H Nishihara: formal analysis, investigation, and visualization.
S Hirakata: conceptualization, supervision, and investigation.
C Owa: formal analysis, validation, and investigation.
S Morishita: supervision, methodology, project administration, and writing—review and editing.
YW Iwasaki: conceptualization, supervision, and project administration.
MC Siomi: conceptualization, supervision, funding acquisition, project administration, and writing—original draft, review, and editing.

## Conflict of Interest Statement

The authors declare that they have no conflict of interest.

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
