## [Reviewer comments · Life Science Alliance]

Flamenco plasticity tunes somatic piRNAs and rewires isoforms, with implications for heritable transposon spread

Mai Moritoh, Chikara Takeuchi, Yurika Namba, Hidenori Nishihara, Shigeki Hirakata, Chie Owa, Shinichi Morishita, Yuka Iwasaki, and Mikiko Siomi

DOI: <https://doi.org/10.26508/lsa.202603650>

Corresponding author(s): Mikiko Siomi, The University of Tokyo

Review Timeline:

Submission Date:	2026-01-30
Editorial Decision:	2026-02-03
Revision Received:	2026-02-28
Editorial Decision:	2026-03-26
Revision Received:	2026-04-02
Accepted:	2026-04-03

Scientific Editor: Tim Fessenden

Transaction Report:

February 3, 2026

Re: Life Science Alliance manuscript #LSA-2026-03650-T

Dr. Mikiko C Siomi
The University of Tokyo
Department of Biological Sciences, Graduate School of Science
Faculty of Science Bldg.3, Rm126
2-11-16 Yayoi
Bunkyo-ku, Tokyo 113-0032
Japan

Dear Dr. Siomi,

Thank you for transferring your manuscript entitled "Flamenco plasticity tunes somatic piRNAs, rewiring isoforms and enabling heritable TE spread" to Life Science Alliance. In agreement with our offer to consider this work, and our subsequent correspondence with you on the previous reviewer requests, we invite you to submit a revised manuscript addressing the following concerns:

- 1) Temper the main claims to reflect that the observations are derived from OSC and not fly ovaries. Here we appreciate that an *in vivo* analysis is hampered due to the "heterogeneous mixture of follicle cells at multiple developmental stages, together with numerous germ cells of large volume" as noted in your prior correspondence. Claims should be adjusted in the abstract, and the inherent limitation of this study should be noted in the discussion. Altering the title is left to your discretion.
- 2) State the ploidy of the OSC line in the text, and adjust claims based on its ploidy status per Reviewers 1 and 2.
- 3) Include additional analysis (in the manner of our choice), to address transcript stability (Reviewer 3) and mRNA transcripts (Reviewer 1) in *Springer*-host hybrids.
- 4) Clarify in the text the source of the ovary piRNAs shown in Figure 3C, per Reviewer 1.

Upon receipt of a revised manuscript according to the above points, we will return this work to the original Reviewer 1 and Reviewer 3 for their evaluation. A point-by-point rebuttal letter should include responses to all reviewer comments in some form, however implementing additional revisions besides those noted above is left to your discretion.

In case you have any questions I would be happy to discuss the revision in more detail via email or phone/videoconferencing. Please let me know which option you prefer, if any.

Thank you for this interesting contribution to Life Science Alliance. We are looking forward to receiving your revised manuscript.

Sincerely,

B. MANUSCRIPT ORGANIZATION AND FORMATTING:

*****IMPORTANT:** It is Life Science Alliance policy that if requested, original data images must be made available. Failure to provide original images upon request will result in unavoidable delays in publication. Please ensure that you have access to all original microscopy and blot data images before submitting your revision. *******

Referee#1

1) *The paper's title, abstract and figure legends all need to include the term "in Drosophila OSC cell lines" relating to Flamenco plasticity and effects from Springer mobilization because the study does not show this Flamenco plasticity or extensive Springer mobilization can happen in the actual Drosophila fly, only in these particular OSC culture lines. The abstract needs more specific and repeated qualifiers that Flamenco cluster structural remodeling and Springer isoforms are only seen in the OSC lines, no sufficient evidence this is possible in actual Drosophila animal follicle cells. The authors should remove the conjecture that there is "relaxation of somatic repression that creates conditions permissive for germ-cell entry" because these genomic aspects of Flamenco and springer are only seen in the OSC lines, not in the actual Drosophila follicle cells.*

We appreciate the referee's comment. To clarify the scope of our observations, we have added "*in cultured OSCs*" where appropriate in the Abstract and Figure legends. We have retained the title as originally written, as suggested by the handling editor at LSA.

With respect to the comment that we should remove the conjecture that "*this relaxation of somatic repression creates conditions permissive for germ-cell entry,*" we addressed this by tempering the wording. The revised text now reads: "*This relaxation of somatic repression may create conditions more permissive for germline entry*" (page 2).

2) *Regarding the capabilities of haplotype-resolution of the OSC genome into just two haplotypes, the results text and analysis need to go into more into depth on how this was achievable if the OSC line is a heterogeneous culture of polyploid and aneuploid segments in these cell culture genomes. Have the authors determined how balanced are all the sequenced regions being represented in the PacBio sequencing to compare to an Illumina WGS to make sure enough of the chromosomes are sufficiently diploid and not too aneuploid to make the haplotype resolution confident? I think this is an important and relevant issue that appears to be too quick to assume is not a factor.*

We appreciate the referee's comment. We performed karyotyping of OSCs used in this study and found that they are diploid (revised Fig S1A).

3) *Pg 3 "Our findings reveal how bursts of TE expression, permitted by Flam remodeling,*

may drive evolutionary novelty." I disagree with this authors' claim from their study, which only has snapshots of the Flam locus at two time points to compare for Flam remodeling, but no other later time point of other OSC lines where Flam remodeling has not occurred where they can show lack of burst of TE expression. Since there is no control example of this, the authors cannot rigorously claim that the effect of a burst of TE expression is caused by Flam remodeling.

We appreciate the referee's comment. In response to this, we have tempered the final paragraph of the Introduction (page 4). Specifically, the last sentence now reads: "transient TE upregulation associated with Flam remodeling could, in some settings, expand isoform diversity and potentially yield evolutionary consequences."

4) Pg 5 "Notably, L(3)mbt-S resembles its vertebrate orthologs (Appendix Figure Fig S1B), supporting the hypothesis that TEs can drive evolutionary innovation." I do not see how this alignment in Fig. S1B between vertebrate orthologs and L(3)mbt-S has any support for the "hypothesis that TEs can drive evolutionary innovation" because that same alignment applies to L(3)mbt-L that forms without the presence of Springer. The reference *Drosophila dm6* genome shows L(3)mbt gene lacking a springer insertion, and I don't recall if neither this paper or the earlier Yamato-Matsuda et al paper ever examined if these two isoforms of L(3)mbt are detected in western blots of *Drosophila* ovaries for a true evolutionary argument to be made? There is also no data to show that L(3)mbt-S has a different or greater activity compared to L(3)mbt-L.

We appreciate the referee's comment and acknowledge that our original explanation was insufficient. Our hypothesis that TEs can promote evolutionary innovation was motivated by the observation that the Springer-driven isoform L(3)mbt-S is more similar to L(3)mbt homolog in other species than the canonical L(3)mbt-L. Without the existence of L(3)mbt-S, the basis for this hypothesis would be less compelling.

With respect to the suggestion to perform Western blotting to detect the two L(3)mbt isoforms in *Drosophila* ovaries: we did not pursue this experiment because we do not have access to flies that naturally (i.e., without genetic manipulation) carry a Springer insertion in ovarian follicle cells on an L(3)mbt allele. Although natural individuals may exist in which a subset of follicle cells harbors the Springer-containing L(3)mbt allele and express L(3)mbt-S, such mosaicism would likely be below the detection limit of Western blotting performed on whole ovaries.

L(3)mbt-S retains transcriptional repressor activity comparable to L(3)mbt-L (original Appendix Fig S1A, retained as Fig S1B in the revised version).

5) Pg 7 "*Additional parameters, Pol II elongation kinetics and acceptor strength, likely modulate this balance.*" *This statement is too speculative to belong in results. It should be moved to the discussion or removed.*

We appreciate the referee's comment. We believe this point warrants only a brief discussion immediately following the relevant results, and we therefore retained it in the revised Results section (page 7). However, to avoid implying that Pol II elongation kinetics and acceptor strength are definitively relevant parameters, we revised the text to "*Additional parameters, such as Pol II elongation kinetics and splice-acceptor strength, may also influence this balance.*"

6) Pg 8 "*This is consistent with a previous report showing that antisense piRNAs in limited amounts are insufficient to trigger silencing (Ariura et al, 2024).*" *This sentence should also cite Post et al 2014 and Genzor et al 2022 as two earlier papers which had first shown that limited antisense piRNAs are insufficient to trigger silencing.*

We appreciate the referee's comment. As suggested, we have now cited two additional papers at the appropriate place (page 8): Post et al., RNA (2014) and Genzor et al., Genome Research (2021).

7) *On the Pg 7-8 and Fig. 2F results of Springer insertion sites into genes to drive hybrid splicing into genes is interesting, but more data that the authors should have needed to be shown particularly in Fig. 2F. The figure should show read support and numbers to evaluate whether the amount of hybrid splicing is meaningful, such as sashimi plots from a STARR alignment of RNAseq to show how much of the springer hybrid splicing is compared to the normal splicing in the genes Rcc1, Sap-r, and Cf2.*

We appreciate the referee's comment. Expression from alleles carrying *Springer* versus those lacking *Springer* will also depend on the relative promoter activities of the *Springer* 5' LTR and the host gene's endogenous promoter. Accordingly, both the magnitude, and potentially even the direction, of the effect are likely to vary across host genes. Consequently, although gene-by-gene comparisons may be informative, it would be difficult to derive a general rule from such individual comparisons.

8) *For all the genes showing in Fig. 2H (both with or without hybrid splicing), there should also be an analysis if the normal gene mRNA is affected or not by the Springer insertion. It is both important and curious to know if the event of hybrid splicing might have a greater effect on the normal mRNA expression than those genes without hybrid splicing. Since the authors make a big deal of the Springer insertions rewiring the transcriptome of the OSCs, the actual impact of the Springer insertions on the normal mRNA expression levels should be quantified. This analysis should also be extended to the genes in Appendix Fig. S4.*

We appreciate the referee's comment. For genes in both the "Hybrid splicing" and "No hybrid splicing" groups (Fig 2H), the canonical mRNA is, in principle, expected to be reduced to approximately half, because only the allele lacking *Springer-intF* can generate the canonical transcript. In genes in the "Hybrid splicing" group, the *Springer-intF*-containing allele instead produces a shorter transcript (lacking the 5' portion relative to the canonical mRNA), and the abundance of this transcript is expected to vary depending on the relative strengths of the host gene promoter and the *Springer* promoter (Therefore, it is difficult to infer a single general consequence of hybrid splicing for canonical mRNA output across all genes). By contrast, in genes in the "No hybrid splicing" group, such as *Marf1*, transcripts from the *Springer-intF*-containing allele were not detected. As discussed in the original manuscript, one likely explanation is poly(A) addition within *Springer-intF* element. Taken together, these observations support our interpretation that intronic *Springer* insertions rewire the OSC transcriptome.

9) *Fig. 3 showing the massive sense piRNA bias to Springer in OSCs is very interesting, and yet there is only one comparison of sense piRNAs to ovaries in Fig. 3C, but there is no detail in the legend if this ovary piRNAs are from dm6 ISO1 flies? Since many of the other figures and analyses are based on the dm6 genome assembly that derived from the ISO1 fly strain that can be requested from the Drosophila stock centers like BDSC and Kyoto DGGR/DGRC, it is imperative the Drosophila ovary small RNAs be analyzed from the ISO1 strain. Buried in the methods, the paper states reanalyzing the published small RNAs from the Shpiz et al 2014 paper that does list a similar genotype as ISO1, but it is not completely definitive. It would be good to show how many Springer-antisense piRNAs are made in the ovary versus the OSC piRNAs.*

We appreciate the referee's comment. We originally cited Shpiz et al. (2014) in the Methods and the paper indicated that the piRNAs were derived from the ISO1 strain. We

have now made this explicit in the revised text (page 8). We compared the proportion of sense-strand *Springer*-derived piRNAs between the ovary (ISO1) and cultured OSCs in the original Fig 3C.

10) The figure legend for Fig. 4 is frustratingly too concise with lacking an explanation of what are the colored bars on the X-scale of the coordinates lines for Fig. 4A, B, and C.

We appreciate the referee's comment, and we apologize that the legend for Fig 4 lacked key details. In response, we have added the necessary explanations to the Fig 4 legend. We have also incorporated the same level of details, where needed, into the legends of other figures.

11) The panel in Appendix Fig. S8B seems more important to be part of the main Fig. 5, because the dot plot of 42AB does show structural variation, but there is sufficient changes to not call this "modest variation" in Pg 11, and I do not agree with the assessment at the bottom of Pg 3: "This plasticity contrasts with the stability of dual-strand clusters in OSCs." The table panel of Fig. S8B does help argue there is less structural variation in 42AB, but I don't fully see this as evidence of stability, I do agree that Flam is a true hotspot of rearrangements in the OSCs.

We appreciate the referee's comment. In this type of dot plot, the extent to which diagonal alignments are preserved provides an overall readout of sequence conservation. For 42AB (Figs 5K and 5L), although the central region shows features suggestive of structural variation, clear diagonal alignments remain visible across substantial flanking regions. We therefore interpreted the overall level of variation in 42AB as relatively limited compared with *Flam*, which shows much more extensive disruption and rearrangement. Regarding the placement of the table panel (Appendix Fig S8B), we kept it in Fig S8 to avoid further increasing the size and complexity of Fig 5.

12) Fig 6 model is very unclear what is trying to be conveyed. What is the WT and the delta-N protein supposed to represent what? The figure legend is much too concise and does not explain what these proteins are in 6C or what is the meaning of the plot in 6A. What is the meaning of the Y-scales in 6A? What is the necessity to depict 6B which only says that the current OSC culture has 2 or 3 more springer insertions than previous cultures.

We appreciate the referee's comment and apologize that the legend for Fig 6 lacked key details. Part of the explanation for the model was provided in the Discussion section of the original text (and remains in the revised text on page 13, where Fig 6C is referred). We have now added a fuller explanation of the model (Fig 6C) to the legend of the corresponding figure. We have also revised the legends for Figs 6A and 6B accordingly.

13) The Fig. 6C model should be removed or completely modified to only depict OSCs genome evolution. The model should not show an intact follicle cell even though OSCs had derived from and share transcriptome profiles with follicle cells, but they have gained extensive polyploidy and aneuploidy so they cannot be interpreted to represent the genomes in intact follicle cells.

We appreciate the referee's comment. The model is intended as a plausible hypothesis that can be derived by integrating the findings of our study. We also note that the diploid nature of cultured OSCs used here is shown in revised Fig S1A.

14) Pg 3 "comparable long-read sequencing" this term is unclear what this means, could be just incorrect grammar. Throughout the text are many other spots where the English grammar is incorrect, so I urge the authors to get assistance on correcting for these grammatical errors, like on Pg 4, first sentence in Results, and Pg 12 what is "decadal sampling"?

We appreciate the referee's comment. We indeed intended to write "comparative" rather than "comparable." English proofreading was performed throughout the revised manuscript.

By "decadal sampling," we mean that we compared OSC datasets collected roughly a decade apart, namely, our current long-read data with previously published OSC short-read datasets from ~10 years earlier, thereby assessing stability and/or changes over an approximately ten-year timescale. We have added this information to the revised text (page 11).

15) Please add line numbers to the manuscript text so that reviewers don't get cranky. And I implore the authors to get all the datasets generated in this study deposited in public repositories immediately, not wait till acceptance of the manuscript. As a reviewer, I would like to see the datasets in a reviewer-token link, this is important regarding which sequencing libraries were made and deposited.

We appreciate the referee's comment. We have added continuous line numbers throughout the revised manuscript. We have also deposited all datasets generated in this study in appropriate public repositories and provide links in the revised Data and materials availability section (page 28).

Referee #2

Major concerns

1) *Insertional bias of Springer and other TEs is confounded by purifying selection and cannot be concluded. To characterize TE's insertional bias, one should minimize selection. However, the passage of OSC over the last 15+ years is letting selection work efficiently, which purges a large number of TE insertions. In particular, authors argue that the tendency to generate "coding sequence-free hybrid splicing" is a feature of Springer, but this (along with the proposed insertional bias) may be the outcome of not just insertion, but also selection and other evolutionary processes.*

We appreciate the referee's comment. Following this reviewer's advice, we modified the corresponding section (page 9), reading, "*We investigated the genomic contexts of Springer insertion (219 Hap_1; 209 in Hap_2) in OSCs. These insertions were preferentially located within a distinctive AT-rich repeat array and displayed a central positional bias within this array (Fig 3E). Genome-wide insertions are enriched in introns relative to exons and intergenic regions (Fig 3F). Notably, although exonic regions comprise 23.8% of the OSC genome (Fig 2D), only 5.9% of Springer insertions fall within exons, substantially below the level expected from motif frequency. The results presented here do not provide definitive evidence for a novel Springer integration bias; rather, they are more consistent with insertion events acquired during OSC establishment and/or subsequent long-term culture.*"

With respect to "coding sequence-free hybrid splicing", we note that this property is consistent with the intrinsic gene architecture of *Springer*. As shown in the original Fig S2A, *Springer* transcripts are spliced to produce Env, and *Springer* exon 1 lacks an AUG start codon; thus, exon 1 does not contribute to the coding sequence. We find that the same principle applies to *Springer*-host hybrid transcripts (e.g., L(3)mbt-S), in which the 5' LTR-derived exon likewise does not add coding sequence.

2) *The argument that the finding of Springer insertions in OSC has implications for*

genetic innovation seems entirely unsupported. OSC is an immortalized cell line, and without any data from follicle cells in vivo, the physiological relevance of these findings in OSC is unclear. In addition, the title calls Springer insertions in OSC "heritable TE spread", but this is based on an extrapolation of OSC findings to follicle cells, and then the prediction that Springer will move to germline and contribute to heritable changes in the genome. In the manuscript, there does not seem to be any evidence that 1) Springer behaves this way in follicle cells, 2) Springer moves to germline, 3) Springer insertion is inherited, 4) Springer insertion contributes to innovation. Thus, the title, abstract, introduction, and discussion may need to refrain from making this point altogether, unless actual data can be presented.

We appreciate the referee's comment. As noted in the original and revised manuscript, "follicle cells encapsulate germ cells" (page 3) throughout nearly the entire course of gametogenesis. Thus, for TEs to reach the germline, they would likely first need to invade in the somatic compartment. Typically, such TE activity is efficiently repressed by the somatic *Flam*-piRNA system (whether established *de novo* or mediated by pre-existing piRNAs). As long as this repression remains effective, opportunities for TEs to gain access to the germline genome should be limited.

Nevertheless, the *Drosophila* germline genome is rich in repetitive TE sequences, suggesting that TE repression is not invariably complete and that transient relaxation may occasionally occur, thereby creating opportunities for germline access.

We acknowledge that we do not currently have direct evidence that this process occurs "naturally" *in vivo*, as ovarian follicle cells are highly heterogeneous and short-lived, making rare events difficult to detect and track. Nonetheless, the well-established TE-rich nature of the germline genome is consistent with our interpretation that fluctuations in somatic repression can, at least on occasion, permit invasion into the germline, which we aim to convey with the model in Fig 6C.

However, following this reviewer's advice, we have tempered our claims throughout the manuscript and revised the text accordingly.

3) The relationship between Flamenco changes and Springer jumping is correlative and not causal. There does not seem to be any experiment that can establish a causal relationship between the changes in Flam and movements of Springer. If so, the title and the abstract are both unsubstantiated.

We appreciate the referee's comment. We agree that our study does not directly

demonstrate causality between *Flam* remodeling and *Springer* movement. However, the remodeling we observe is not restricted to *Springer*-related sequences; instead, it extends across broader regions of *Flam*, suggesting a locus-wide restructuring that reshapes the somatic piRNA landscape and is therefore plausibly linked to changes in *Springer* repression. However, following this reviewer's advice, we have revised the Abstract accordingly. We have retained the title as originally written, as suggested by the handling editor at LSA.

4) *It is unclear to me whether OSC is diploid or polyploid. Throughout the manuscript, authors talk about two haplotypes, but do we know the ploidy of OSC genome? Related to this point, what is the heterozygosity of OSC? Given that many TE insertions are recessive and would be invisible to selection, how does this knowledge of heterozygosity affect conclusions throughout the manuscript regarding insertion bias and the fitness effects of Springer insertions?*

We appreciate the referee's comment. We performed karyotyping and found that OSCs used in this study are diploid (revised Fig S1A).

5) *It seems that intronic Springer insertions in the opposite orientation of host genes could also alter gene expression in interesting ways. Have authors done unbiased analysis (not restricted to intronic sense insertions) to conclude that only the intronic sense insertions show interesting behaviors as reported in this version of the manuscript?*

We appreciate the referee's comment. We are currently investigating how intronic *Springer* insertions in the opposite orientation affect host gene expression, and we anticipate that these findings will be reported in future work.

6) *If I understand it correctly, dual strand piRNA clusters and 20A are only expressed in germline and not in OSC. If so, technically they are not piRNA producing loci in OSC—they are more like inert heterochromatin—so they are not a meaningful comparison against flamenco. To argue that flamenco is special, perhaps an unbiased genome-wide analysis is warranted.*

We appreciate the referee's comment. To highlight that *Flam* is unusual in this respect, we analyzed conservation across the X chromosome (Figs 5D–F). These data show that, although the X chromosome is broadly well conserved, the *Flam* locus stands out as a

region of reduced conservation (indicated by the small open box in each panel). Enlarged views of the boxed regions are shown in Figs 5A–C.

Because the direct comparison between *Flam* and dual-strand clusters/20A (original Figs 5 and S8B–F) is informative, we retain this analysis in the revised manuscript. However, we concluded that cross-cluster comparison of the ATAC-seq data would not be sufficiently informative and therefore removed original Fig S9.

Minor points

1) *I am not sure if I missed the recent developments in the field, but isn't 20A a uni-strand cluster? The manuscript referred to it as dual strand for some reason.*

We apologize for our oversight. We have now corrected this issue throughout the manuscript where appropriate.

2) *I can be wrong, but I thought Gypsy1 and Springer have almost identical internal sequences. Could authors include the TE consensus sequences in the supplement?*

Indeed, the body regions of *Springer* and *gypsy1* (6,739 nt and 6,732 nt, respectively) are highly similar (99.67% sequence identity). However, when a 7-nt gap is considered, the two elements differ by approximately 30 nt overall, with these differences scattered across the body region. In addition, their LTR sequences are distinct from one another (5'+ 3', 807 nt and 986 nt, respectively). Consistent with these differences, *Springer* and *gypsy1* are annotated as separate TEs in FlyBase.

In this study, we first performed genome analyses focused on full-length and near-full-length *Springer* insertions (Fig 2), which required *Springer* and *gypsy1* to be treated as distinct elements. We also treated them separately in Fig S5F, where full-length and near-full-length *gypsy1* insertions in the OSC genome were analyzed.

In other analyses, such as those shown in Fig 4, RepeatMasker (v4.1.5) (Smit et al, 2015) was used, and these two TEs were likewise annotated separately, including at the level of fragmented sequences. In Fig S6, piRNA reads mapping to regions shared between the *Springer* and *gypsy1* body sequences were probabilistically assigned by random multimapping. Nevertheless, we confirmed that some piRNA peaks were specific to either *Springer* or *gypsy1*, reflecting single-nucleotide differences between the two elements.

We regret that we did not include the TE consensus sequences in the manuscript. Because the body regions of *Springer* and *gypsy1* are highly similar, as

noted above, we instead focused on analyzing them as separately annotated elements.

3) *The first mention of Fig 3D in text could better explain what "enrichment" means.*

We have added a clearer explanation of what we mean by "enrichment" at the point where Fig 3D is first introduced in the revised text (page 8).

4) *It might be helpful to be consistent with using T or U when describing motifs in text.*

We apologize for our oversight. We have now corrected this issue throughout the manuscript where appropriate.

5) *It seems to me that there is no need to invoke "horizontal transfer" of TEs at the third paragraph of Introduction. It is irrelevant to the story and came out of nowhere.*

We have removed the term "horizontal transfer" from the third paragraph of the revised Introduction (page 3).

6) *Caught a couple grammatical errors: "ley" is not a correct word, and the word "and" is missing in the sentence "we generated a haplotype-resolved OSC genome by PacBio HiFi long-read sequencing integrated the data with Micro-C XL profiles".*

We apologize for our oversight. We have corrected these issues and carefully checked the manuscript to ensure that no additional grammatical errors were overlooked.

Referee #3

Major concerns

1) *Fate of Springer-host hybrid transcripts: The authors identified numerous Springer-driven isoforms initiated from the 5' LTR and spliced into the host exon. For L(3)mbt, this results in an N-terminally truncated protein. However, it is less clear whether this is a general feature.*

1-1) *Do all (or most) identified hybrid transcripts produce translatable mRNAs? Clarifying this point is important for supporting the idea that Springer insertions drive functional innovation rather than the events merely being tolerated.*

We thank the referee for this important question. As shown in the original Table S3, one of the 72 genes carrying a Springer-intF insertion (CR44885 in Hap_1) is annotated as an lncRNA. We therefore excluded this case and examined the remaining 71 cases,

focusing on the host-derived regions of the corresponding transcripts. We found that all 71 host-derived regions contain one or more AUG codons, indicating that these transcripts are, in principle, “translatable”. However, in response to this referee’s comment, we have tempered the language throughout the manuscript where appropriate.

1-2) Are there examples where the hybrid transcript isoform is non-coding or predicted to be unstable?

We thank the referee for this important question. This point is closely related to the concern above. As noted there, CR44885 in Hap_1 is annotated as an lncRNA. Excluding this case, we examined the remaining 71 host-derived regions and found that all contain one or more AUG codons, suggesting that these transcripts are not, in principle, non-coding RNAs.

Regarding transcript stability, our current data do not indicate that *Springer*-driven hybrid splicing is intrinsically associated with instability. For example, L(3)mbt-S is expressed at a level readily detectable by Western blotting and was reported to be even more abundant than the canonical L(3)mbt-L (Yamamoto-Matsuda et al, 2022). In the present study, mCherry (Figs 1C and 1D) and truncated Piwi (Fig S9) were also robustly detected at levels comparable to their corresponding controls. Together, these observations support the view that *Springer*-driven hybrid splicing does not inherently generate unstable products. Stability may also vary depending on the host gene context.

1-3) If feasible, a brief categorization (e.g., predicted translated vs likely non-translated) of the 72 identified Springer-driven isoforms would help interpret their functional impact.

We appreciate the referee’s helpful suggestion. As described above, we classified the 71 *Springer*-intF-containing genes (except lncRNA:CR44885) as predicted translatable, because each contains one or more AUG codons within the host-derived region. We further grouped these cases into five categories based on the relationship between the *Springer*-driven transcript and the canonical isoform (revised Table S3): (a) alternative TSS, canonical TIS (Het); (b) predicted N-terminal truncation (Het); (c) predicted out-of-frame aberrant protein (Het); (d) predicted biallelic N-terminal truncation; and (e) predicted biallelic aberrant protein. The abbreviations used in revised Table S3 are as follows: TSS, transcription start site; TIS, translation initiation site; Het, a heterozygous *Springer*-intF insertion (one allele carries *Springer*-intF, whereas the other does not); and

“biallelic,” indicating that both alleles carry a *Springer-intF* insertion. We hope that this revision satisfactorily addresses the referee’s concern.

2) *OSCs versus fly ovaries: The described mechanism appears specific to OSCs and is not evident in the dm6 reference genome or fly ovaries.*

2-1) *Can the authors comment on why this Springer-Flam relationship is observed in OSCs but not in vivo?*

We appreciate the referee’s insightful comment. We think that the *Springer-Flam* relationship is inherently difficult to detect directly under physiological conditions. *In vivo*, follicle cells are embedded in a three-dimensional tissue environment and are highly heterogeneous across developmental stages, and bulk ovary sample also contain abundant, large germ cells. In this context, even if *Flam* remodeling occurs in only one or a small subset of follicle cells, such events would likely fall below the detection limit in bulk tissue analyses. Cultured OSCs provide an *ex vivo* system with a simplified and relatively stable cellular context that is difficult to recapitulate *in vivo*. Accordingly, we interpret the *Springer-Flam* relationship as a mechanistic insight revealed by cultured OSCs, while acknowledging that comparable events may occur at substantially lower frequency and/or be much less readily detectable under physiological conditions.

2-2) *To what extent might this reflect properties of cultured OSCs (e.g., altered chromatin accessibility, different selection pressure, absence of germline signaling)?*

We appreciate the referee’s important comment. *In vivo*, follicle cells are embedded in a three-dimensional tissue environment (the ovary) and, as development proceeds, undergo extensive interactions with neighboring germ cells (and likely with other follicle cells). Over developmental time, their transcriptional states, chromatin organization, and signaling responses change dynamically, making the population inherently heterogeneous. In contrast, cultured OSCs exist in a greatly simplified *ex vivo* context in which much of this environmental input is absent; consequently, they can be maintained over long-term passage while remaining relatively homogeneous. Accordingly, we consider OSCs to be an especially favorable system for identifying and analyzing relationship such as the *Springer (TE)-Flam* relationship described in this study. A similar process could, in principle, occur in follicle cells; however, even if such events arise in only a small subset of cells, they would likely fall below the detection limit in bulk analyses

of a heterogeneous tissue. Such rare events could still be biologically consequential. Transient relaxation of somatic TE repression in a limited number of follicle cells could create an opportunity for otherwise silenced TEs to become active and, potentially, access neighboring germ cells, where they might enter the germline genome and be transmitted to the next generation. This is the scenario we envision based on the insights gained in this study, and it is illustrated in the model in Fig 6C.

2-3) *Could interaction with germ cells or the germline piRNA pathway constrain similar events in intact ovaries?*

This is indeed an interesting question to address. However, we regret that we cannot provide a definitive answer to this question at present.

3) *Persistence of non-silencing sense piRNAs from flamenco in OSCs: The authors show that flamenco remodeling leads to the production of predominantly sense Springer piRNAs that do not efficiently silence the element.*

3-1) *Do the authors envision this as a transient "maladaptive" state, or as a stable outcome of ongoing cluster remodeling?*

We thank the referee for raising this interesting question. At present, we favor the possibility that the current state is transient rather than a stable endpoint, and we speculate that further *Flam* remodeling could eventually generate antisense *Springer* piRNAs. If such a state were to arise in one or a small number of OSCs, it could, in principle, trigger more robust silencing of *Springer* elements genome-wide, including intronic copies. However, it remains unclear whether such a state would be compatible with OSC viability or instead impose a fitness cost. We attempted to ectopically induce antisense *Springer* piRNAs using the system described in our previous study (Ishizu et al. *Cell Reports* 2015; doi:10.1016/j.celrep.2015.06.035). In our hands, however, the amount of piRNAs produced appeared insufficient to achieve global *Springer* silencing, and we therefore were unable to obtain conclusive results from this experiment.

3-2) *Is there evidence that such sense piRNA production is tolerated because it is selectively neutral in OSCs, or actively maintained by some mechanism?*

We thank the referee for raising this question. At present, we do not have evidence that allows us to distinguish between these possibilities, namely, whether the sense piRNA production is tolerated because it is largely selectively neutral in OSCs, or whether it is

actively maintained by some mechanism. Although we cannot exclude the possibility of active maintenance, we currently see no clear functional rationale for such a mechanism. We therefore currently favor, as a working interpretation, that this sense-biased piRNA production reflects a tolerated state in cultured OSCs.

4) *Some discussion of how such "non-beneficial" piRNA populations persist would help clarify the model.*

We appreciate this suggestion. We propose that “*non-beneficial*” piRNA populations may persist as “*by-products*” of piRNA biogenesis and, if they impose little fitness cost, may be subject to only weak negative selection in cultured OSCs. We have added this point to the revised Discussion section (page 13).

Minor concerns

1) *Appendix Fig. S3: The legend does not provide sufficient details. Please clarify what the reporter assay measures as well as relevant experimental information for data interpretation.*

We apologize that the necessary information was missing. We have now added a clear description of the reporter assay, along with the relevant experimental details needed to interpret the data (e.g., Fig S3 legend).

2) *Appendix Fig. S5A: The meaning of "relative expression" is unclear. Is this a ratio of normalized transcript abundance in Piwi-depleted versus control OSCs?*

As the referee noted, we meant “*a ratio of normalized transcript abundance in Piwi-depleted versus control OSCs.*” We have now added this wording to the legend of Fig S5A.

3) *Appendix Fig. S8C-F: A color key is missing.*

We added a color key to Figs S8C–F. Thank you.

4) *Appendix Fig. S10: The legend needs more details. The experimental setup and the cell types used should be explicitly stated.*

We have now added the experimental setup and the cell type used to the legend of revised Fig S9. Thank you.

March 26, 2026

RE: Life Science Alliance Manuscript #LSA-2026-03650-TR

Dr. Mikiko C Siomi
The University of Tokyo
Department of Biological Sciences, Graduate School of Science
Faculty of Science Bldg.3, Rm126
2-11-16 Yayoi
Bunkyo-ku, Tokyo 113-0032
Japan

Dear Dr. Siomi,

Thank you for submitting your revised manuscript entitled "Flamenco plasticity tunes somatic piRNAs, rewiring isoforms and enabling heritable transposon spread". This work was returned to the original Reviewers 1 and 3 for their evaluation. As you will see, these reviewers diverged in their appraisal of this work. Reviewer 1 remains concerned that findings in OSCs do not warrant the (now revised) claims on Flam and piRNAs in vivo. They also note the evidence for OSC diploid status is not convincing, which also relates to their concern on expression levels (prior point 8). Reviewer 3 is supportive with no major concerns, but makes several important suggestions to improve the clarity of these findings in particular by more carefully rewording the figure legends.

We sincerely appreciate the detail and care taken by Reviewer 1, and we agree that the conclusions must align with the evidence shown. In view of their concerns, we suggest changing the title to "Flamenco plasticity tunes somatic piRNAs and rewires isoforms, with implications for heritable transposon spread". Regarding the diploid status of OSCs, the discussion section must acknowledge this remains a limitation of the present study in the absence of more rigorous/quantitative measurement of their ploidy. Finally, we invite you to consider improving the legend for Figure 4 sought by Reviewer 1 and the several important suggestions raised by Reviewer 3 to improve the text. We would be happy to publish your paper in Life Science Alliance pending these revisions, as well as final changes necessary to meet our formatting guidelines.

MANUSCRIPT ORGANIZATION AND FORMATTING:

To avoid unnecessary delays in the acceptance and publication of your paper, please read the following information carefully. Full guidelines are available on our Instructions for Authors page, <https://www.life-science-alliance.org/authors>

- Please add an ORCID ID for the corresponding author - you should have received instructions on how to do so.
- Please add a Summary Blurb/Alternate Abstract in our system.
- Please add the X and Bluesky handles of your host institute/organization, as well as your own, and/or one of the authors, in our system.
- The contributions selected for Shinichi Morishita do not qualify them for authorship. Please either update the contributions in our system and in the Author Contributions section of the manuscript, or let us know if the authors need to be removed (and added potentially to the acknowledgment section).

We welcome submissions of potential cover images for the issue of LSA in which your work would appear. If you have high quality images associated with this work, please feel free to email these, with a caption, to the journal office.

LSA encourages authors to provide a 30-60 second video where the study is briefly explained. We will use these videos on

social media to promote the published paper and the presenting author (for examples, see <https://docs.google.com/document/d/1-UWCfbE4pGcDdcgzcmiuJl2XMBJnxKYeqRvLLrLSo8s/edit?usp=sharing>). Corresponding or first-authors are welcome to submit the video. Please submit only one video per manuscript. The video can be emailed to contact@life-science-alliance.org

FINAL FILES:

The following items are required for acceptance.

The license to publish form must be signed before your manuscript can be sent to production. A link to the license to publish form will be available to the corresponding author only. Please take a moment to check your funder requirements.

Thank you for your attention to these final processing requirements. Please revise and format the manuscript and upload materials as soon as you are able.

Thank you for this interesting contribution to the literature. We look forward to publishing your paper in Life Science Alliance.

Sincerely,

Reviewer #1 (Comments to the Authors (Required)):

Overall, I am quite unsatisfied with the many unresolved issues with this revision and the authors' rebuttal letter. Although the data on Flamenco changes being linked to the transposon composition changes in the OSC cell line are clear in this study, the other statements in the paper still make much too strong and unsupported for improperly extending the claims of genome rewiring and heritability to the fly germline. The study basically just examines two time point isolates of the OSC line where there has been a spontaneous change in the structure of the Flam locus as well as expansion of new Springer insertions, but no directed and guided changes to Flamenco like mutations, so the causal connections cannot be substantiated by the authors current data.

I still disagree on the paper title not stating "in Drosophila OSC cell lines" with the rest of the title claiming that the Flamenco locus does the following: "[1] tunes somatic piRNAs, [2]rewiring isoforms and [3] enabling heritable transposon spread" for which there is zero evidence for the later two by the actual Flam locus in the Drosophila animal. For the point of "rewiring isoforms", this term is unclear for what "isoforms" are (are these gene or mRNA or TE isoforms) and this paper is unclear as to whether the Flam piRNA cluster is the entity doing this or if it is really the Springer transposon mobilization events that are doing this. For the point of "enabling heritable transposon spread", again there is no data in the fly that mutations or perturbations to Flam's plasticity directly leads to transposon changes in progeny, because of the known severe Flam mutations, the females are sterile. This study only performs long-read sequencing of OSC genomes after prolonged culture, and discover what are spontaneous and selected major structural rearrangements to Flam in the OSC, but no such direct change in the fly. Therefore, I cannot concur with the authors rebuttal and the LSA handling editor claim that the original title be retained. I still cannot endorse the

original paper title.

The introduction text makes references to TEs being "vertically transmitted across generations" and "TEs requiring germline establishment" and "colonize[ing] the germline" at Lines 73 to 75; but all of these events can only truly be assessed in actual animal reproduction, which this study completely lacks any data from the fly, only has data from the cultured OSCs. The cultured OSCs are very useful for biochemistry and cell biology aspects of cell culture inheritance, but the authors must refine the language in this manuscript to remove all of these other referrals to actual animal genetics.

For example, the new text in the abstract at Lines 39-41 "relaxation of somatic repression may create conditions more permissive for germline entry, raising the possibility of heritable genome change." is completely inappropriate and unsupported for this claim that implies genetic effects in the fly animal when this study only has data from the OSC cells. All of these findings are just in the OSC cells, and that part is fine, but it is inappropriate to then extend these conclusions to the animal, which cannot be pushed by these authors who only show data from the OSC cell culture systems.

The new image in Fig. S1A is insufficient for the authors to claim karyotyping of just one N=1 image to claim their OSC line is a diploid genome. This is an extremely shoddy response based on expediency, shockingly from the strong reputation of rigor from the famous Mikiko Siomi lab. The authors should look at Fig. S1A of Sytnikova et al Genome Res 2014 where they show a much better karyotyping analysis of OSC and OSS cell lines. The Fig S1A of Sytnikova et al shows N=72 and N=61 for OSC and OSS cell images, respectively, and while 79% of the OSC cells are diploid, 21% are polyploid!

Because I cannot believe from this insufficient reply by these authors for the one single image of claimed karyotyping, my original concern for how the polyploidy was unresolved in the long read sequencing, and thus still remains unaddressed for Points #2 and #12 and #13 of my original critique.

The authors response to my Point #7 on the lack of read support to back up Figure 2F is a dodge to my request and completely do-able. If they used RNAseq data to make claims of hybrid splicing between Springer and the Rcc1, Sap-r and Cf2 mRNAs, the authors can create an artificial mini-genome of these gene models with the Springer insertion to run STARR or use a TE insertion genomics program like TEMP from Zhipeng Weng's lab to count how many reads actually support the hybrid splicing transcript. The reason this request is biologically important is because this figure is arguing for significance in transcriptome "rewiring" by these Springer TE hybrid transcripts, but if the reads support for these TE hybrid transcripts are less than 10% or less than 1% of the normally spliced exons, then the true biological context of these hybrid splicing events will be more evident to the audience and not just hyped up by the authors. For the authors to claim that it is difficult to derive a general rule from such individual comparisons is dodging the issue I raised, which is to just do and show the analysis for those individual genes Rcc1, Sap-r and Cf2!

Similarly, the authors response to my Point #8 is another dodge of my request. The authors' rebuttal does not show any new analysis in the revision nor in the rebuttal letter, only an assertion that "the canonical mRNA is, in principle, expected to be reduced to approximately half, because only the allele lacking Springer-intF can generate the canonical transcript." Where is the actual analysis to show this? There should be an analysis on each gene's expression that has a Springer-intF insertion that can be compared between OSC Hap_1 and OSC_Hap2, where one cell line has the Springer-intF and the other cell line lacks it, and gene expression can be compared. The big claim of transcriptome rewiring needs more quantitative read count support, not just show these diagrams of the hybrid gene structure and supplying an unsatisfactory response that lacks new analyses to support the response.

The Legend for Figure 4 is improved and I can finally understand what all the colors representing what in the diagrams. However, something that is still missing from this long read resequencing of Flamenco is what sequence element is used to anchor all these Flam assemblies to define the "0.0" sequence anchor point? Is it the nearby DIP1 gene upstream of the Flam TSS? What about the downstream locus beyond Flam? These details need to be spelled out enable the reader to judge the validity of Fig 4D as to how consistent are the synteny lines being drawn.

The Legend for Figure 6 is improved, but the figure still has the egregious depiction of follicle cells and claims of Flam remodeling in follicle cells leading to a release of TE expression and transfer from the follicle cell to the oocyte. This model is completely unsupported by no true genetics experiments in fly ovaries that are missing from this study or any other study! The OSC line is a far-removed cell culture system and by no way can be construed to be a surrogate for the actual follicle cell biology in the ovary. I cannot understand the obstinance of the authors to stand by this claim and again their so-called "karyotype analysis" in Fig. S1A is completely insufficient. The model would only be acceptable to highlight genome dynamics in the OSC cell line by itself as an entity in cell culture, but any reference to follicle cells and oocytes in the ovary must be removed because this study lacks any true genetic data to support this claim.

Reviewer #2 (Comments to the Authors (Required)):

The study addresses how a horizontally transmitted LTR transposon, Springer, evades somatic piRNA-mediated repression in

cultured *Drosophila* ovarian somatic cells (OSCs) while reshaping host gene isoforms. Using long-read sequencing analyses, the authors show that Springer inserts into promoter-proximal introns, where its 5'LTR initiates transcription that splices into downstream host exons without contributing coding sequence. In parallel, the major somatic piRNA cluster, Flamenco, undergoes structural remodeling, leading to mainly sense piRNAs that fail to silence Springer. The findings provide a specific example of how the somatic piRNA pathway may be retuned to permit TE persistence and potential germline entry.

Overall, most claims are substantiated by the data. The study is technically strong, well described, and conceptually interesting. The authors have addressed most of my earlier conceptual questions and those arising from insufficient experimental details, which clarified several points. It remains unclear to what extent the authors consider the possibility that OSCs harbor culture-specific features that are not present in vivo. I do not see OSC-specific features as a weakness as long as the conclusions are not overstated, since the methodologies and proposed model remain valuable. That said, while the revised manuscript softens direct claims about in vivo relevance, and the OSCs have historically served as an excellent model for in vivo piRNA pathway studies, the overall presentation still makes it easy to interpret the results as directly reflecting physiological mechanisms. This limitation has been noted by the other reviewers. For the sake of usefulness, I focus here on more general and presentation-related concerns.

Major Comments

1. In multiple figures, the legends tend to emphasize conclusions or interpretations rather than clearly describing the experiments or analyses performed. Many technical details are provided in the Methods, but it is often difficult to map those descriptions to specific figures. This reduces the readability of the data. Overall, figure legends should provide sufficient experimental context and avoid presenting conclusions without sufficient methodological context.

Some examples:

- Figure 1 (line 926): The abbreviations "FL" and "LS" should be explicitly defined in the legend.
- Figure 2H (lines 946-948): The legend presents a conclusion rather than describing the analysis, but the conclusion itself seems partially speculative in the main text. The legend should clearly state what is plotted and define what each dot represents. A minor but useful change would be including the p-value directly on the graph.
- Figure 3: The legends mostly describe interpretations rather than the underlying analyses. The interpretations per se are fine, but they are a little confusing without the general experimental descriptions.
- Figure 3D (Line 957): The legend states enrichment without explaining how enrichment was calculated or what background or baseline was used.
- Figure 3A: "Piwi-bound piRNAs" is mentioned, but it is not clear whether this refers to Piwi-IP data or how many replicates were used.
- Figure S2D (Line 1039): The experiment is not fully explained. The figure appears to show exogenous expression of Springer in S2 cells, but this is not explicitly stated in the text or legend.
- Figure S3B-D (Lines 1043): Although annotations are added, the experimental setup remains unclear. For example, what was the reporter construct? How was it expressed? How was RNA analyzed/sequenced? The main text only briefly refers to a "genomic reporter," and additional plasmid construct details are buried in the Methods. A concise and coherent description should be included in the legend or main text.
- Figure S9: Similarly, the experimental design is not immediately clear. The reader must infer details from multiple sections, which should not be necessary.

2. There are several instances where wording overstates what is directly supported by the data:

- Line 89: "...71 additional Springer-driven isoforms, underscoring its broad mutagenic potential." The term "broad" may not be fully justified given that the baseline expectation is unclear.
- Figure 1B: The transcription and translation sites should be labeled as "predicted" sites.
- Lines 168-176: This paragraph is mostly speculative. While it is interesting, the section might be clearer if merged with the preceding section.

3. There are several instances of an imprecise choice of words, which reduced the clarity. For example:

- Line 922: Unless I misunderstood the meaning, "dm6" should be positioned later in the sentence.
- Line 1008: "genomic-seq" is not standard terminology.
- Line 1014: "Model" is too vague for a figure title here, especially since it is a speculative view of the in vivo pathway. Perhaps consider "A proposed model of...."
- Line 193: The sentence feels somewhat incomplete. The intended meaning (enrichment of Springer insertions) is not clearly conveyed.
- Figure 3 (Line 950): "enriched in" instead of "rich in."

Minor comments

- Figure S7: I suggest either removing sense vs. antisense information or including them consistently across all panels.
- Figure (Line 972): Sp#1 and Sp#2 are explained in the main text but should also be briefly annotated in the figure legend for better readability.

Point-to-point response

1) *We sincerely appreciate the detail and care taken by Reviewer 1, and we agree that the conclusions must align with the evidence shown. In view of their concerns, we suggest changing the title to "Flamenco plasticity tunes somatic piRNAs and rewires isoforms, with implications for heritable transposon spread".*

We thank the editor for this helpful suggestion. In accordance with it, we have revised the title to: *"Flamenco plasticity tunes somatic piRNAs and rewires isoforms, with implications for heritable transposon spread."*

2) *Regarding the diploid status of OSCs, the discussion section must acknowledge this remains a limitation of the present study in the absence of more rigorous/quantitative measurement of their ploidy.*

We thank the editor for this helpful suggestion. We recently performed quantitative measurement of the diploid status of OSCs (n = 30) and found that 93.3% were diploid. We have now included this result in the figure legend of Fig S1A. Fig S1A continues to show a single representative image, consistent with Figure S1 of Sytnikova et al. (2014). Because the ploidy issue was addressed experimentally, we did not include additional discussion of it in the Discussion section.

3) *Finally, we invite you to consider improving the legend for Figure 4 sought by Reviewer 1 and the several important suggestions raised by Reviewer 3 to improve the text.*

We thank the editor for these thoughtful and specific suggestions. In response, we have revised the legend of Fig 4, as requested by Reviewer 1, and have also incorporated the important suggestions raised by original Reviewer 3 (currently Reviewer 2) to improve the text. Please see the "Point-to-point response" below for details.

Point-to-point response**Reviewer 1**

- *Point #7: The Legend for Figure 4 is improved and I can finally understand what all the colors representing what in the diagrams. However, something that is still missing from this long read resequencing of Flamenco is what sequence element is used to anchor all these Flam assemblies to define the "0.0" sequence anchor point? Is it the nearby DIP1 gene upstream of the Flam TSS? What about the downstream locus beyond Flam? These details need to be spelled out enable the reader to judge the*

validity of Fig 4D as to how consistent are the synteny lines being drawn.

We thank the reviewer for pointing this out. As described in the original Methods, we defined the *Flam* locus as the genomic region between CG32820 and CG14621, which were identified as conserved flanking genes in both the dm6 reference annotation and our lifted-over OSC gene annotations. The validity of this definition is supported by RepeatMasker annotation, which showed that this region is enriched for TE fragments relative to the surrounding genomic regions, and by the fact that piRNAs map to this region. Accordingly, the 5' boundary of the *Flam* locus, defined by these conserved flanking genes, was used as the 0.0 anchor point in Figs 4A–D. We have revised the figure legends accordingly.

Reviewer 2

Major Comments

1. In multiple figures, the legends tend to emphasize conclusions or interpretations rather than clearly describing the experiments or analyses performed. Many technical details are provided in the Methods, but it is often difficult to map those descriptions to specific figures. This reduces the readability of the data. Overall, figure legends should provide sufficient experimental context and avoid presenting conclusions without sufficient methodological context.

Some examples:

- Figure 1 (line 926): The abbreviations "FL" and "LS" should be explicitly defined in the legend.

We thank the reviewer for this point. We now added following wording in the legend of Fig 1C: "FL-LS, full-length L3-Springer; mCh, mCherry."

- Figure 2H (lines 946-948): The legend presents a conclusion rather than describing the analysis, but the conclusion itself seems partially speculative in the main text. The legend should clearly state what is plotted and define what each dot represents. A minor but useful change would be including the p-value directly on the graph.

We thank the reviewer for this point. We agree that the original legend to Fig 2H was overly interpretive. We have revised the legend to clarify that each dot represents a single *Springer* insertion and that the plot shows the log₁₀-transformed distance (bp) to the end of the nearest downstream exon, grouped according to the presence or absence of detected hybrid splicing. We also added the p-value directly to the graph.

- Figure 3: The legends mostly describe interpretations rather than the underlying

analyses. The interpretations per se are fine, but they are a little confusing without the general experimental descriptions.

- Figure 3D (Line 957): The legend states enrichment without explaining how enrichment was calculated or what background or baseline was used.

We thank the reviewer for this point. For Fig 3D, we now explicitly define the enrichment metric and clarify that the dm6 reference genome was used as the baseline for comparison. These revisions make the legend more informative and reduce ambiguity.

- Figure 3A: "Piwi-bound piRNAs" is mentioned, but it is not clear whether this refers to Piwi-IP data or how many replicates were used.

We thank the reviewer for this point. In Fig 3A, we reanalyzed Piwi-bound piRNA data obtained by immunoprecipitation and reported by Murano et al. (2019) (see lines 180–181 of the original text), which is now cited also in the figure legend. Readers interested in experimental details are referred to the original paper.

- Figure S2D (Line 1039): The experiment is not fully explained. The figure appears to show exogenous expression of Springer in S2 cells, but this is not explicitly stated in the text or legend.

We thank the reviewer for pointing this out and apologize for the lack of clarity. We have now added the following sentence to the legend of Fig S2D: *"The L3-Springer element was introduced into S2 cells and ectopically expressed, and the resulting transcripts were examined for polyadenylation. All cDNA clones analyzed (n = 10) contained poly(A) tails."*

- Figure S3B-D (Lines 1043): Although annotations are added, the experimental setup remains unclear. For example, what was the reporter construct? How was it expressed? How was RNA analyzed/sequenced? The main text only briefly refers to a "genomic reporter," and additional plasmid construct details are buried in the Methods. A concise and coherent description should be included in the legend or main text.

We thank the reviewer for this suggestion. We have revised the legend of Fig S3 accordingly.

- Figure S9: Similarly, the experimental design is not immediately clear. The reader must infer details from multiple sections, which should not be necessary.

We thank the reviewer for this point. In accordance with it, we have revised the legend of Fig S9.

2. There are several instances where wording overstates what is directly supported by the data:

- Line 89: "...71 additional Springer-driven isoforms, underscoring its broad mutagenic potential." The term "broad" may not be fully justified given that the baseline expectation is unclear.

We thank the reviewer for this point. We have revised the wording to "underscoring its mutagenic potential."

- Figure 1B: The transcription and translation sites should be labeled as "predicted" sites.

We thank the reviewer for this point. In our earlier study (Yamamoto-Matsuda et al, 2022), we determined the TSS and the TIS. We were unable to distinguish between Met311 and Met325 as the TIS. We have added the following sentence to the figure legend: "The TSS was determined experimentally, whereas the TIS was estimated to be either Met311 or Met325 in our previous study (Yamamoto-Matsuda et al, 2022)."

- Lines 168-176: This paragraph is mostly speculative. While it is interesting, the section might be clearer if merged with the preceding section.

We thank the reviewer for this point. In accordance with this suggestion, we have merged this paragraph into the preceding section.

3. There are several instances of an imprecise choice of words, which reduced the clarity. For example:

- Line 922: Unless I misunderstood the meaning, "dm6" should be positioned later in the sentence.

We thank the reviewer for this point. We apologize that the original wording was unclear and have revised the sentence as follows: "Structures of L(3)mbt in the dm6 genome (upper) and Hap_2 in the OSC genome (lower)."

- Line 1008: "genomic-seq" is not standard terminology.

We thank the reviewer for this point. We have revised "genomic-seq" to "genomic sequencing."

- Line 1014: "Model" is too vague for a figure title here, especially since it is a speculative view of the in vivo pathway. Perhaps consider "A proposed model of..."

We thank the reviewer for this point. We have revised "Model" to "A proposed model for how Flam plasticity shapes host–transposon interactions."

- *Line 193: The sentence feels somewhat incomplete. The intended meaning (enrichment of Springer insertions) is not clearly conveyed.*

We thank the reviewer for this point. To clarify the intended meaning, we have revised the sentence as follows: “*Springer showed the strongest copy-number enrichment in the OSC genome, increasing by ~36-fold relative to the dm6 reference (Fig 3D).*”

- *Figure 3 (Line 950): "enriched in" instead of "rich in."*

We have revised the wording accordingly. Thank you for pointing this out.

Minor comments

- *Figure S7: I suggest either removing sense vs. antisense information or including them consistently across all panels.*

We have added sense/antisense information to all panels in Figs S6 and S7. Thank you for pointing this out.

- *Figure (Line 972): Sp#1 and Sp#2 are explained in the main text but should also be briefly annotated in the figure legend for better readability.*

We thank the reviewer for this point. We have added the following sentence in the legend of Fig 4A: “*Notably, the two Springer fragments, Sp#1 and Sp#2 in the dm6 genome (pale green), and four gypsy1 fragments, gy1#1-4 (pale purple), are oriented opposite (reverse) to the direction of Flam transcription. Springer-targeting piRNAs map to Sp#1 and Sp#2 (Fig S6A), indicating that these regions serve as sources of Springer-targeting piRNAs.*”

April 3, 2026

RE: Life Science Alliance Manuscript #LSA-2026-03650-TRR

Dr. Mikiko C Siomi
The University of Tokyo
Department of Biological Science Graduate School of Science University of Tokyo
Faculty of Science Bldg.3, Rm126 2-11-16 Yayoi, Bunkyo-ku
Tokyo 113-0032
Japan

Dear Dr. Siomi,

Thank you for submitting your Research Article entitled "Flamenco plasticity tunes somatic piRNAs and rewires isoforms, with implications for heritable transposon spread". It is a pleasure to let you know that your manuscript is now accepted for publication in Life Science Alliance. Congratulations on this interesting work, and many thanks for your diligence and responsiveness during the revision process. As was noted in our prior correspondence, please ensure that the methods are updated during the proofing process.

DISTRIBUTION OF MATERIALS:

Again, congratulations on a very nice paper. I hope you are pleased with how the manuscript was handled editorially. We look forward to future exciting submissions from your lab.

Sincerely,
